# Unbounded Differentially Private Quantile and Maximum Estimation

**David Durfee**
Anonym Inc.
david@anonymco.com

## Abstract

In this work we consider the problem of differentially private computation of
quantiles for the data, especially the highest quantiles such as maximum, but with
an unbounded range for the dataset. We show that this can be done efficiently
through a simple invocation of `AboveThreshold`, a subroutine that is iteratively
called in the fundamental Sparse Vector Technique, even when there is no upper
bound on the data. In particular, we show that this procedure can give more
accurate and robust estimates on the highest quantiles with applications towards
clipping that is essential for differentially private sum and mean estimation. In ad-
dition, we show how two invocations can handle the fully unbounded data setting.
Within our study, we show that an improved analysis of `AboveThreshold` can
improve the privacy guarantees for the widely used Sparse Vector Technique that
is of independent interest. We give a more general characterization of privacy loss
for `AboveThreshold` which we immediately apply to our method for improved
privacy guarantees. Our algorithm only requires one $O(n)$ pass through the data,
which can be unsorted, and each subsequent query takes $O(1)$ time. We empiri-
cally compare our unbounded algorithm with the state-of-the-art algorithms in
the bounded setting. For inner quantiles, we find that our method often performs
better on non-synthetic datasets. For the maximal quantiles, which we apply
to differentially private sum computation, we find that our method performs
significantly better.

## 1   Introduction

In statistics, quantiles are values that divide the data into specific proportions, such as median
that divides the data in half. Quantiles are a central statistical method for better understanding a
dataset. However, releasing quantile values could leak information about specific individuals within
a sensitive dataset. As a result, it becomes necessary to ensure that individual privacy is ensured
within this computation. Differential privacy offers a rigorous method for measuring the amount
that one individual can change the output of a computation. Due it's rigorous guarantees, differential
privacy has become the gold standard for measuring privacy. This measurement method then offers
an inherent tradeoff between accuracy and privacy with outputs of pure noise achieving perfect
privacy. Thus, the goal of designing algorithms for differentially private quantile computation is to
maximize accuracy for a given level of privacy.

There are a variety of previous methods for computing a given quantile of the dataset that we will
cover in Section 1.2 , but each of these requires known bounds on the dataset. The most effective
and practical method invokes the exponential mechanism Smith (2011). For computing multiple
quantiles this method can be called iteratively. Follow-up work showed that it could be called
recursively by splitting the dataset at each call to reduce the privacy cost of composition Kaplan
et al. (2022). Further, a generalization can be called efficiently in one shot  Gillenwater et al. (2021).

## 1.1 Our contributions

In this work, we offer an alternative practical and accurate approach, Unbounded Quantile Estimation (UQE), that also invokes a well-known technique and can additionally be applied to the unbounded setting. While the commonly-used technique designs a distribution to draw from that is specific to the dataset, our method will simply perform a noisy guess-and-check. Initially we assume there is only a lower bound on the data, as non-negative data is common in real world datasets with sensitive individual information. Our method will simply iteratively increase the candidate value by a small percentage and halt when the number of data points below the value exceeds the desired amount dictated by the given quantile. While the relative increase will be small each iteration, the exponential nature still implies that the candidate value will become massive within a reasonable number of iterations. As a consequence, our algorithm can handle the unbounded setting where we also show that two calls to this procedure can handle fully unbounded data. Computing multiple quantiles can be achieved by applying the recursive splitting framework from Kaplan et al. (2022).

Performing our guess-and-check procedure with differential privacy exactly fits `AboveThreshold`, a method that is iteratively called in the Sparse Vector Technique Dwork et al. (2009). We also take a deeper look at `AboveThreshold` and unsurprisingly show that similar to report noisy max algorithms, the noise addition can come from the Laplace, Gumbel or Exponential distributions. We further push this analysis to show that for monotonic queries, a common class of queries which we will also utilize in our methods, the privacy bounds for composition within the Sparse Vector Technique can be further improved. Given the widespread usage of this technique,[1] we believe this result is of independent interest. Furthermore, we give a more general characterization of query properties that can improve the privacy bounds of `AboveThreshold`. We immediately utilize this characterization in our unbounded quantile estimation algorithm to improve privacy guarantees.

While the commonly used algorithms for quantile estimation can still apply incredibly loose bounds to ensure the data is contained within, this can have a substantial impact upon the accuracy for estimating the highest quantiles such as maximum. This leads to an especially important application for our algorithm, differentially private sum computation, which can thereby be used to compute mean as well. Performing this computation practically without assumptions upon the distribution often requires clipping the data and adding noise proportionally. Clipping too high adds too much noise, and clipping too low changes the sum of the data too much. The highest quantiles of the data are used for clipping to optimize this tradeoff. The unbounded nature of our approach fundamentally allows us to estimate the highest quantiles more robustly and improve the accuracy of differentially private sum computation.

This improvement in differentially private sum computation is further evidenced by our empirical evaluation, with significant improvements in accuracy. Our emperical comparison will be upon the same datasets from previous work in the bounded setting. We also compare private computation of the inner quantiles on these datasets. For synthetic datasets generated from uniform or guassian distributions, we see that the more structured approach of designing a distribution for the data from the exponential mechanism consistently performs better. However, for the real-world datasets, we see that our unstructured approach tends to perform better even within this bounded setting. By design our algorithm is less specific to the data, so our alternative approach becomes advantageous when less is known about the structure and bounds of the data *a priori*. As such, for large-scale privacy systems that provide statistical analysis for a wide variety of datasets, our methods will be more flexible to handle greater generality accurately.

## 1.2 Background literature

The primary algorithm for privately computing a given quantile, by which we compare our technique, applies the exponential mechanism with a utility function based upon closeness to the true quantile Smith (2011). We will discuss this algorithm, which we denote as Exponential Mechanism Quantile (EMQ), in greater detail in Appendix A. This approach was then extended to computing multiple quantiles more cleverly by recursively splitting the data and establishing that only one partition of

---

[1]For example, see Roth and Roughgarden (2010); Hardt and Rothblum (2010); Dwork et al. (2015); Nissim et al. (2016); Nissim and Stemmer (2018); Kaplan et al. (2020a,b); Hasidim et al. (2020); Bun et al. (2017); Bassily et al. (2018); Cummings et al. (2020); Ligett et al. (2017); Barthe et al. (2016a,b); Steinke and Ullman (2016); Cummings et al. (2015); Ullman (2015); Nandi and Bassily (2020); Shokri and Shmatikov (2015); Hsu et al. (2013); Sajed and Sheffet (2019); Feldman and Steinke (2017); Blum et al. (2015); Chen et al. (2016)

the dataset can change between neighbors thereby reducing the composition costs Kaplan et al. (2022). Additional follow-up work showed that a generalization of the utility function to multiple quantiles could be efficiently drawn upon in one shot Gillenwater et al. (2021). Another recent result examined this problem in the streaming data setting and gave a method that only uses strongly sub-linear space complexity Alabi et al. (2022).

Quantile computation can also be achieved through CDF estimation Bun et al. (2015); Kaplan et al. (2020a). However these techniques offer limited practicality as they rely upon several reductions and parameter tuning. Recursively splitting the data is also done for CDF estimation algorithms where the statistics from each split can be aggregated for quantile computation Dwork et al. (2010); Chan et al. (2011). These techniques tend to be overkill for quantile estimation and thus suffer in accuracy comparatively.

We will also give improved privacy analysis of the Sparse Vector Technique which was originally introduced in Dwork et al. (2009). A more detailed analysis of the method can be found in Lyu et al. (2017). Additional recent work has shown that more information can be output from the method at no additional privacy cost Kaplan et al. (2021); Ding et al. (2023).

### 1.3 Organization

We provide the requisite notation and definitions in Section 2. In Section 3, we review the `AboveThreshold` algorithm from the literature and show that privacy analysis can be further improved. In Section 4, we provide our unbounded quantile estimation method. In Section 5, we test our method compared to the previous techniques on synthetic and real world datasets. In Appendix A, we consider the estimation of the highest quantiles which has immediate application to differentially private sum and mean estimation. In Appendix B, we give further results on the `AboveThreshold` algorithm and provide the missing proofs from Section 3. In Appendix C, we provide further variants and extensions of our unbounded quantile estimation technique.

## 2 Preliminaries

We will let $x, x'$ denote datasets in our data universe $\mathcal{X}$.

**Definition 2.1.** *Datasets $x, x' \in \mathcal{X}$ are neighboring if at most one individual's data has been changed.*

Note that we use the *swap* definition, but our analysis of the `AboveThreshold`algorithm will be agnostic to the definition of neighboring. Using this definition as opposed to the *add-subtract* definition is necessary to apply the same experimental setup as in Gillenwater et al. (2021). Our differentially private quantile estimation will apply to either and we will give the privacy guarantees if we instead use the *add-subtract* definition in Appendix C.2.

**Definition 2.2.** *A function $f : \mathcal{X} \to \mathbb{R}$ has sensitivity $\Delta$ if for any neighboring datasets $|f(x) - f(x')| \leq \Delta$*

**Definition 2.3.** *Dwork et al. (2006b,a) A mechanism $M : \mathcal{X} \to \mathcal{Y}$ is $(\varepsilon, \delta)$-differentially-private (DP) if for any neighboring datasets $x, x' \in \mathcal{X}$ and $S \subseteq \mathcal{Y}$:*

$$\Pr[M(x) \in S] \leqslant e^{\varepsilon}\Pr[M(x') \in S] + \delta.$$

We will primarily work with pure differential privacy in this work where $\delta = 0$. We will also be considering the composition properties of the Sparse Vector Technique, and the primary method for comparison will be Concentrated Differential Privacy that has become widely used in practice due to it's tighter and simpler advanced composition properties Bun and Steinke (2016). This definition is instead based upon Reny divergence where for probability distributions $P, Q$ over the same domain and $\alpha > 1$

$$D_{\alpha}(P\|Q) = \frac{1}{\alpha - 1} \ln \mathop{\mathbf{E}}_{z \sim P} \left[ \left( \frac{P(z)}{Q(z)} \right)^{\alpha-1} \right]$$

**Definition 2.4.** *Bun and Steinke (2016) A mechanism $M : \mathcal{X} \to \mathcal{Y}$ is $\rho$-zero-concentrated-differentially-private (zCDP) if for any neighboring datasets $x, x' \in \mathcal{X}$ and all $\alpha \in (1, \infty)$:*

$$D_{\alpha}(M(x)\|M(x')) \leq \rho\alpha.$$

We can translate DP into zCDP in the following way.

**Proposition 1.** *Bun and Steinke (2016) If $M$ satisfies $\varepsilon$-DP then $M$ satisfies $\frac{1}{2}\varepsilon^2$-zCDP*

In our examination of `AboveThreshold` we will add different types of noise, similar to the report noisy max algorithms Ding et al. (2021). Accordingly, we will consider noise from the Laplace, Gumbel and Exponential distributions where $\mathsf{Lap}(b)$ has PDF $p_{\mathsf{Lap}}(z; b)$, $\mathsf{Gumbel}(b)$ has PDF $p_{\mathsf{Gumbel}}(z; b)$, and $\mathsf{Expo}(b)$ has PDF $p_{\mathsf{Expo}}(z; b)$ where

$$p_{\mathsf{Lap}}(z; b) = \frac{1}{2b} \exp\left(-|z|/b\right) \qquad\qquad p_{\mathsf{Gumbel}}(z; b) = \frac{1}{b} \exp\left(-\left(z/b + e^{-z/b}\right)\right)$$

$$p_{\mathsf{Expo}}(z; b) = \begin{cases} \frac{1}{b} \exp\left(-z/b\right) & z \geq 0 \\ 0 & z < 0 \end{cases}$$

We let $\mathsf{Noise}(b)$ denote noise addition from any of $\mathsf{Lap}(b), \mathsf{Gumbel}(b)$, or $\mathsf{Expo}(b)$. We will also utilize the definition of the exponential mechanism to analyze the addition of $\mathsf{Gumbel}$ noise.

**Definition 2.5.** *McSherry and Talwar (2007) The Exponential Mechanism is a randomized mapping $M : \mathcal{X} \rightarrow \mathcal{Y}$ such that*

$$Pr\left[M(x) = y\right] \propto \exp\left(\frac{\varepsilon \cdot q(x, y)}{2\Delta}\right)$$

*where $q : \mathcal{X} \times \mathcal{Y} \rightarrow \mathbb{R}$ has sensitivity $\Delta$.*

## 3 Improved Analysis for Sparse Vector Technique

In this section, we review the `AboveThreshold` algorithm from the literature. To our knowledge, this technique has only been used with Laplace noise in the literature. Unsurprisingly, we show that Gumbel and Exponential noise can also be applied, with the former allowing for a closed form expression of each output probability. We further show that for monotonic queries the privacy analysis of the Sparse Vector Technique, which iteratively applies `AboveThreshold`, can be improved. All proofs are be pushed to Appendix B where we also give a more general characterization of query properties that can improve the privacy bounds of `AboveThreshold`.

### 3.1 Above Threshold Algorithm

We first provide the algorithm for `AboveThreshold` where noise can be applied from any of the Laplace, Gumbel or Exponential distributions.

---
**Algorithm 1** `AboveThreshold`

---
**Require:** Input dataset $x$, a stream of queries $\{f_i : \mathcal{X} \rightarrow \mathbb{R}\}$ with sensitivity $\Delta$, and a threshold $T$
 1: Set $\hat{T} = T + \mathsf{Noise}(\Delta/\varepsilon_1)$
 2: **for** each query $i$ **do**
 3:      Set $v_i = \mathsf{Noise}(\Delta/\varepsilon_2)$
 4:      **if** $f_i(x) + v_i \geq \hat{T}$ **then**
 5:          Output $\top$ and `halt`
 6:      **else**
 7:          Output $\bot$
 8:      **end if**
 9: **end for**

---

We will also define a common class of queries within the literature that is often seen to achieve a factor of 2 improvement in privacy bounds.

**Definition 3.1.** *We say that stream of queries $\{f_i : \mathcal{X} \rightarrow \mathbb{R}\}$ with sensitivity $\Delta$ is monotonic if for any neighboring $x, x' \in \mathcal{X}$ we have either $f_i(x) \leq f_i(x')$ for all $i$ or $f_i(x) \geq f_i(x')$ for all $i$.*

To our knowledge all previous derivations of `AboveThreshold` in the literature apply `Lap` noise which gives the following privacy guarantees.

**Lemma 3.1** (Theorem 2 and 3 of Lyu et al. (2017)). *If the noise addition is `Lap` then Algorithm 1 is $(\varepsilon_1 + 2\varepsilon_2)$-DP for general queries and is $(\varepsilon_1 + \varepsilon_2)$-DP for monotonic queries.*

Given that `Expo` noise is one-sided `Lap` noise, it can often be applied for comparative algorithms such as this one and report noisy max as well. We will show this extension in the appendix for completeness.

**Corollary 3.1.** *If the noise addition is `Expo` then Algorithm 1 is $(\varepsilon_1 + 2\varepsilon_2)$-DP for general queries and $(\varepsilon_1 + \varepsilon_2)$-DP for monotonic queries.*

While the proofs for `Expo` noise generally follow from the `Lap` noise proofs, it will require different techniques to show that `Gumbel` noise can be applied as well. In particular, we utilize the known connection between adding `Gumbel` noise and the exponential mechanism.

**Lemma 3.2.** *If the noise addition is `Gumbel` and $\varepsilon_1 = \varepsilon_2$ then Algorithm 1 is $(\varepsilon_1 + 2\varepsilon_2)$-DP for general queries and $(\varepsilon_1 + \varepsilon_2)$-DP for monotonic queries.*

We defer the proof of this to the appendix. In all of our empirical evaluations we will use `Expo` noise in our calls to `AboveThreshold` because it has the lowest variance for the same parameter. While we strongly believe that `Expo` noise will be most accurate under the same noise parameters, we leave a more rigorous examination to future work. We also note that this examination was implicitly done for report noisy max between `Gumbel` and `Expo` noise in McKenna and Sheldon (2020), where their algorithm is equivalent to adding `Expo` noise Ding et al. (2021), and `Expo` noise was shown to be clearly superior.

### 3.2 Improved privacy analysis for Sparse Vector Technique

In this section, we further consider the iterative application of `AboveThreshold` which is known as the sparse vector technique. We show that for monotonic queries, we can improve the privacy analysis of sparse vector technique to obtain better utility for the same level of privacy. Our primary metric for measuring privacy through composition will be zCDP which we defined in Section 2 and has become commonly used particularly due to the composition properties. We further show in the appendix that our analysis also enjoys improvement under the standard definition of differential privacy. These improved properties immediately apply to our unbounded quantile estimation algorithm as our queries will be monotonic.

**Theorem 1.** *If the queries are monotonic, then for any noise addition of `Lap`, `Gumbel`, or `Expo` we have that Algorithm 1 is $\frac{1}{2}\varepsilon^2$-zCDP where $\varepsilon = \frac{\varepsilon_1}{2} + \varepsilon_2$. If the noise addition is `Gumbel` then we further require $\varepsilon_1 = \varepsilon_2$*

Note that applying Proposition 1 will instead give $\varepsilon = \varepsilon_1 + \varepsilon_2$. It will require further techniques to reduce this by $\varepsilon_1/2$ which will immediately allow for better utility with the same privacy guarantees. This bound also follows the intuitive factor of 2 improvement that is often expected for monotonic queries. The analysis will be achieved through providing a range-bounded property, a definition that was introduced in Durfee and Rogers (2019). This definition is ideally suited to characterizing the privacy loss of selection algorithms, by which we can view `AboveThreshold`. As such it will also enjoy the improved composition bounds upon the standard differential privacy definition shown in Dong et al. (2020). This range bounded property was then unified with zCDP with improved privacy guarantees in Cesar and Rogers (2021).

We give a proof of this theorem along with further discussion in Appendix B.3. We will also give a generalized characterization of when we can take advantage of properties of the queries to tighten the privacy bounds in `AboveThreshold`. We will immediately utilize this characterization to improve the privacy guarantees for our method in Appendix C.

## 4 Unbounded Quantile Estimation

In this section we give our method for unbounded quantile estimation. We focus upon the lower bounded setting which we view as most applicable to real-world problems, where non-negative

data is incredibly common, particularly for datasets that contain information about individuals. This method can be symmetrically apply to upper bounded data, and in the appendix we will show how this approach can be extended to the fully unbounded setting.

For the quantile problem we will assume our data $x \in \mathbb{R}^n$. Given quantile $q \in [0, 1]$ and dataset $x \in \mathbb{R}^n$ the goal is to find $t \in \mathbb{R}$ such that $|\{x_j \in x | x_j < t\}|$ is as close to $qn$ as possible.

## 4.1 Unbounded quantile mechanim

The idea behind unbounded quantile estimation will be a simple guess-and-check method that will just invoke AboveThreshold. In particular, we will guess candidate values $t$ such that $|\{x_j \in x | x_j < t\}|$ is close to $qn$. We begin with the smallest candidate, recall that we assume lower bounded data here and generalize later, and iteratively increase by a small percentage. At each iteration we check if $|\{x_j \in x | x_j < t\}|$ has exceeded $qn$, and terminate when it does, outputting the most recent candidate. In order to achieve this procedure privately, we will simply invoke AboveThreshold. We also discuss how this procedure can be achieved efficiently in Section 4.2.

Thus we give our algorithm here where the lower bound of the data, $\ell$, is our starting candidate and $\beta$ is the scale at which the threshold increases. Given that we want our candidate value to increase by a small percentage it must start as a positive value. As such we will essentially just shift the data such that the lower bound is instead 1. In all our experiments we set $\beta = 1.001$, so the increase is by 0.1% each iteration.

---

**Algorithm 2** Unbounded quantile mechanism

---

**Require:** Input dataset $x$, a quantile $q$, a lower bound $\ell$, and parameter $\beta > 1$
  1: Run AboveThreshold with $x$, $T = qn$ and $f_i(x) = |\{x_j \in x | x_j - \ell + 1 < \beta^i\}|$
  2: Output $\beta^k + \ell - 1$ where $k$ is the query that AboveThreshold halted at

---

Given that our method simply calls AboveThreshold it will enjoy all the privacy guarantees from Section 3.1. Furthermore we will show that our queries are monotonic.

**Lemma 4.1.** *For any sequence of thresholds $\{t_i \in \mathbb{R}\}$ let $f_i(x) = |\{x_j \in x | x_j < t_i\}|$ for all i. For any neighboring dataset under Definition 2.1, we have that $\{f_i\}$ are monotonic queries with sensitivity 1.*

*Proof.* Let $x_j$ be the value that differs between neighbors $x, x'$. Define $S_{x,t} = \{x_j \in x | x_j < t\}$. We consider the case $x_j' > x_j$ and the other will follow symmetrically. For all thresholds $t_i \in (-\infty, x_j]$ we have $x_j \notin S_{x,t_i}$ and $x_j' \notin S_{x',t_i}$, so $f_i(x) = f_i(x')$. For all thresholds $t_i \in (x_j', \infty)$ we have $x_j \in S_{x,t_i}$ and $x_j' \in S_{x',t_i}$, so $f_i(x) = f_i(x')$. Finally, for all thresholds $t_i \in (x_j, x_j']$ we have $x_j \in S_{x,t_i}$ and $x_j' \notin S_{x',t_i}$, so $f_i(x) = f_i(x') + 1$. Therefore, $f_i(x) \geq f_i(x')$ for all $i$, and the sensitivity is 1. $\square$

Note that for the *swap* definition of neighboring the threshold remains constant. We will discuss how to extend our algorithm and further improve the privacy bounds for the *add-subtract* definition of neighboring in the appendix.

## 4.2 Simple and scalable implementation

In this section we show how our call to AboveThreshold can be done with a simple linear time pass through the data, and each subsequent query takes $O(1)$ time. While the running time could potentially be infinite, if we set $\beta = 1.001$, then after 50,000 iterations our threshold is already over $10^{21}$ and thus highly likely to have halted. Unless the scale of the data is absurdly high or the $\beta$ value chosen converges to 1, our guess-and-check process will finish reasonably quickly. [2]

In our initial pass through the data, for each data point $x_j$ we will find the index $i$ such that $\beta^i \leq x_j - \ell + 1 < \beta^{i+1}$, which can be done by simply computing $\lfloor \log_\beta(x_j - \ell + 1) \rfloor$ as our lower bound ensures $x_j - \ell + 1 \geq 1$. Using a dictionary or similar data structure we can efficiently store $|\{x_j \in x | \beta^i \leq x_j - \ell + 1 < \beta^{i+1}\}|$ for each $i$ with the default being 0. This preprocessing does not

---

[2]Note that the process can also be terminated at any time without affecting the privacy guarantees.

require sorted data and takes $O(n)$ arithmetic time, where we note that the previous algorithms also measure runtime arithmetically.

Finally, for each query if we already have $|\{x_j \in x | x_j - \ell + 1 < \beta^i\}|$, then we can add $|\{x_j \in x | \beta^i \leq x_j - \ell + 1 < \beta^{i+1}\}|$ in O(1) time to get $|\{x_j \in x | x_j - \ell + 1 < \beta^{i+1}\}|$. Inductively, each query will take O(1) time. We provide the code in the appendix for easier reproducibility.

### 4.3 Extension to multiple quantiles

The framework for computing multiple quantiles set up in Kaplan et al. (2022) is agnostic to the technique used for computing a single quantile. Their method will first compute the middle quantile and split the data according to the result. Through recursive application the number of levels of computation will be logarithmic. Furthermore, at each level we can see that at most one partition of the data will differ between neighbors, allowing for instead a logarithmic number of compositions. As such our approach can easily be applied to this recursive splitting framework to achieve the same improvements in composition. This will require some minor updating of their proofs to the *swap* definition that we will do in the appendix.

## 5 Empirical Evaluation

In this section we empirically evaluate our approach compared to the previous approaches. We give further detail of the previous approaches, particularly EMQ, in Appendix A along with strong intuition upon why our approach will better handle maximal quantile estimation for data clipping. We first go over the datasets and settings used for our experiments which will follow recent related work Gillenwater et al. (2021); Kaplan et al. (2022). Next we evaluate how accurately our method estimates quantiles for the different datasets in the bounded setting. Finally, we will consider the application of computing differentially private sum, which also gives mean computation, and show how our algorithm allows for a significantly more robust and accurate method when tight bounds are not known for the dataset.

### 5.1 Datasets

We borrow the same setup and datasets as Gillenwater et al. (2021); Kaplan et al. (2022). We test our algorithm compared to the state-of-the-art on six different datasets. Two datasets will be synthetic. One draws 10,000 data points from the uniform distribution in the range $[-5, 5]$ and the other draws 10,000 data points from the normal distribution with mean zero and standard deviation of five. Two datasets will come from Soumik (2019) with 11,123 data points, where one has book ratings and the other has book page counts. Two datasets will come from Dua and Graf (2019) with 48,842 data points, where one has the number of hours worked per week and the other has the age for different people. We provide histograms of our datasets for better understanding in Figure 1.

### 5.2 Quantile estimation experiments

For our quantile estimation experiments, for a given quantile $q \in [0, 1]$ we consider the error of outcome $o_q$ from one of the private methods to be $|o_q - t_q|$ where $t_q$ is the true quantile value. We use the in-built quantile function in the numpy library with the default settings to get the true quantile value. As in previous related works, we randomly sample 1000 datapoints from each dataset and run the quantile computation on each method. This process is then iterated upon 100 times and the error is averaged. We set $\varepsilon = 1$ as in the previous works, which will require setting $\varepsilon_1 = \varepsilon_2 = 1/2$ for the call to AboveThreshold in our method.

We will also tighten the ranges to the following, $[-5, 5]$ for the uniform dataset, $[-25, 25]$ for the normal dataset, $[0, 10]$ for the ratings dataset, $[0, 10000]$ for the pages dataset, $[0, 100]$ for the hours dataset, and $[0, 100]$ for the ages dataset. Given that EMQ suffers performance when many datapoints are equal we add small independent noise to our non-sythetic datasets. This noise will be from the normal distribution with standard deviation 0.001 for the ratings dataset and 0.1 for the other three that have integer values. Our method does not require the noise addition but we will use the perturbed dataset for fair comparison. True quantiles are still computed upon the original data. For the datasets with integer values we rounded each output to the nearest integer. For our method we set $\beta = 1.001$ for all datasets.

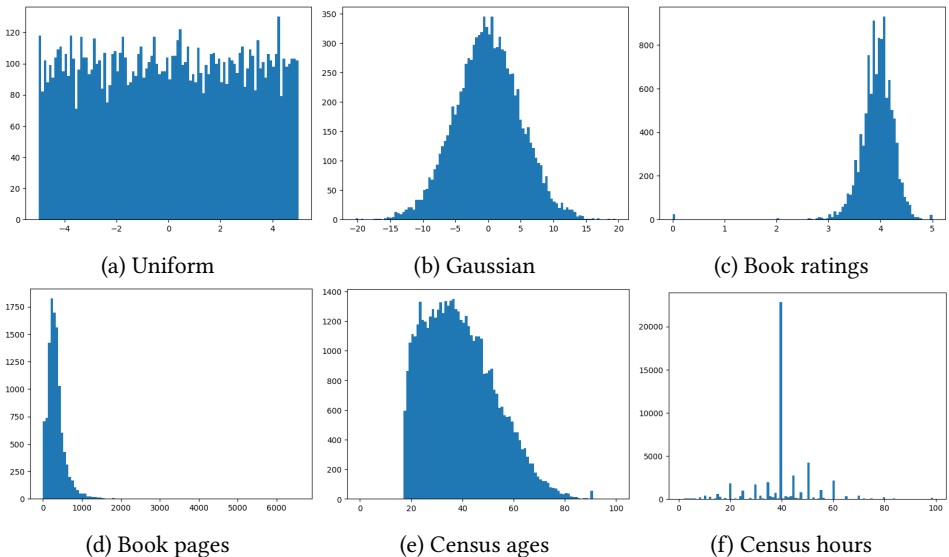

(a) Uniform      (b) Gaussian      (c) Book ratings

(d) Book pages      (e) Census ages      (f) Census hours

Figure 1: Histograms for each of our datasets.

For these experiments we only compare our method UQE and the previous method EMQ, using the implementations from Gillenwater et al. (2021). The other procedures, discussed in the appendix are more generalized and thus for this specific setting do not perform nearly as well which can be seen in the previous experiments Gillenwater et al. (2021); Kaplan et al. (2022), so we omit them from our results. For this experiment we consider estimating each quantile from 5% to 95% at a 1% interval. In Figure 2 we plot the mean absolute error of each normalized by the mean absolute error of UQE to make for an easier visualization.

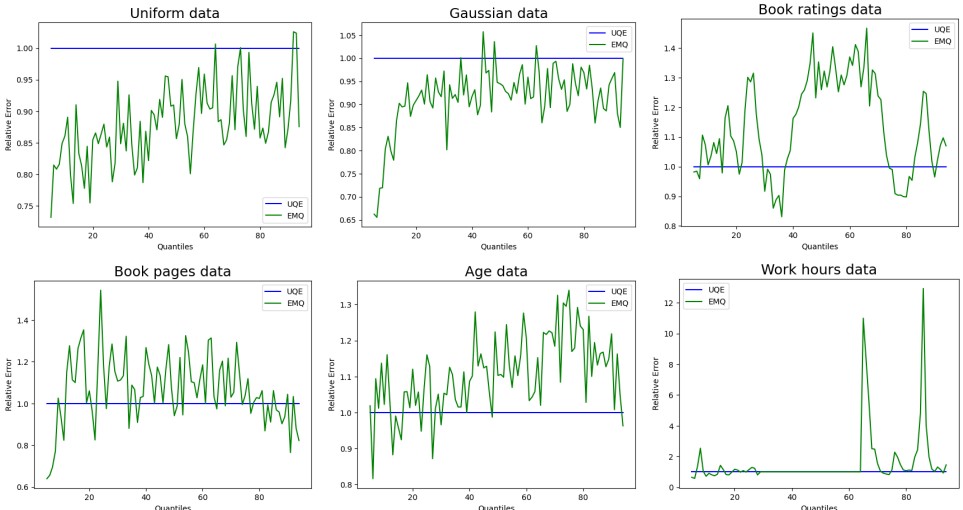

Figure 2: Plots of UQE = (mean absolute error UQE) / (mean absolute error UQE) and EMQ = (mean absolute error EMQ) / (mean absolute error UQE). Normalizing in this way will make for an easier visualization. When EMQ is below UQE then it's error is lower, and when EMQ is above UQE then it's error is higher

As we can see in Figure 2, EMQ consistently performs better on synthetic data, and UQE tends to performs better on the non-synthetic data. This fits with our intuition that UQE will be best suited to situations where the data is unstructured and less is known about the dataset beforehand because our guess-and-check methodology is designed to better handle ill-behaving datasets.

## 5.3 Sum estimation experiments

As the primary application of our method we will also be considering differentially private sum computation, which can thereby compute mean as well. We will be using the following $2\varepsilon$-DP general procedure for computing the sum of non-negative data:

1. Let $\Delta = \text{PrivateQuantile}(x, q, \varepsilon)$ where $\text{PrivateQuantile}$ is any differentially private computation algorithm and $q \approx 1$.
2. Output $\text{Lap}(\Delta/\varepsilon) + \sum_{j=1}^{n} \min(x_j, \Delta)$

We further test this upon the non-synthetic datasets. For large-scale privacy systems that provide statistical analysis for a wide variety of datasets, if we use a $\text{PrivateQuantile}$ that requires an upper bound then we ideally want this bound to be agnostic to the dataset. The is particularly true for sum computations upon groupby queries as the range and size can differ substantially amongst groups. As such, we fix the range at $[0, 10000]$ to encompass all the datasets. We will otherwise use the same general setup as in Section 5.2.

We will measure the error of this procedure as the absolute error of the output and the true sum of the dataset. For each of the 100 iterations of choosing 1000 samples randomly from the full dataset, we also add $\text{Lap}$ noise 100 times. Averaging over all these iterations gives our mean absolute error.

| Privacy | Method | Ratings data | Pages data | Ages data | Work hours data |
|---------|--------|--------------|------------|-----------|-----------------|
| | UQE | $4.78_{0.21}$ | $4385.23_{2077.16}$ | $103.05_{16.04}$ | $180.48_{44.92}$ |
| $\varepsilon = 1$ | EMQ | $5.73_{0.54}$ | $4324.38_{2343.25}$ | $187.06_{33.55}$ | $339.06_{75.82}$ |
| | AT | $8.75_{0.31}$ | $4377.45_{2340.93}$ | $293.11_{117.32}$ | $471.98_{174.07}$ |
| | UQE | $9.22_{0.31}$ | $7102.34_{3093.13}$ | $180.61_{27.03}$ | $277.89_{77.60}$ |
| $\varepsilon = 0.5$ | EMQ | $6906.13_{7123.36}$ | $7601.47_{3963.52}$ | $678.22_{1931.11}$ | $2131.80_{4669.11}$ |
| | AT | $29.01_{115.04}$ | $7491.97_{4367.31}$ | $473.19_{238.19}$ | $582.58_{369.29}$ |
| | UQE | $44.59_{1.79}$ | $21916.37_{6423.75}$ | $821.77_{157.68}$ | $981.10_{219.66}$ |
| $\varepsilon = 0.1$ | EMQ | $45861.79_{28366.46}$ | $46552.09_{27944.18}$ | $50558.32_{28843.72}$ | $47185.58_{29437.22}$ |
| | AT | $11351.77_{21423.40}$ | $30830.49_{20422.05}$ | $14490.06_{25268.71}$ | $9928.18_{20159.17}$ |

Table 1: Mean absolute error for differentially private sum estimation. The standard deviation over the 100 iterations is also provided for each in the subscript. UQE = Our unbounded quantile estimation method. EMQ = The exponential mechanism based quantile estimation method. AT = The aggregate tree method for quantile estimation. For our method we only use $q = 0.99$. For the others we use the best performance for $q \in \{0.95, 0.96, 0.97, 0.98, 0.99\}$.

We will run this procedure with $\varepsilon \in \{0.1, 0.5, 1\}$. Further we will use $q = 0.99$ always for our method, but give the absolute error for the best performing $q \in \{0.95, 0.96, 0.97, 0.98, 0.99\}$ for the other methods. It is important to note that this value would have to be chosen ahead of time, which would add more error to the other methods. The previous methods we consider here are again the EMQ, but also the aggregate tree (AT) methods, were we use both the implementation along with generally best performing height (3) and branching factor (10) from Gillenwater et al. (2021). We also implemented the bounding technique using inner quartile range within Algorithm 1 of Smith (2011), but this performed notably worse than the others so we omitted the results from our table.

As we can see in Table 1, our method is far more robust and accurate. Furthermore, for our method the choice of $q$ remained constant and we can see that our results still stayed consistently accurate when $\varepsilon$ changed. Note that the noise added to the clipped sum is also scaled proportional to $\varepsilon$ so the amount the error increased as $\varepsilon$ decreased for our method is what would be expected proportionally. Once again these findings are consistent with our intuition. Our technique is more robust to differing datasets and privacy parameters, and especially better performing for this important use case.

Recall that the sampled data had size 1000 so dividing accordingly can give the error on mean estimates. There is a long line of literature on differentially private mean estimation.[3] To our knowledge, all of these more complex algorithms either require assumptions upon the data distribution,

---

[3]See Kamath et al. (2022) for an extensive review of this literature

such as sub-Gaussian or bounded moments, or bounds upon the data range or related parameters, and most often require both. These results also focus upon proving strong theoretical guarantees of accuracy with respect to asymptotic sample complexity. We first note that our approach will provide better initial bounds upon the data as seen in our experiments, which directly improve the theoretical guarantees in the results that require a data range. But also our focus here is upon practical methods that are agnostic to data distributions and more widely applicable to real-world data. Consequently, a rigorous comparison among all of these methods would be untenable and outside the scope of this work.

### 5.4 Parameter tuning

We kept our $\beta$ parameter fixed in all experiments for consistency but also to make our method data agnostic. However, our choice was aggressively small in order to achieve higher precision in the inner quantile estimation comparison in Section 5.2. This choice was still highly resilient to changes in $\varepsilon$ for our sum experiments as we see our error only scaled proportional to the increase in noise. But for more significant decreases in $\varepsilon$ or in the data size, i.e. the conditions under which all private algorithms suffer substantial accuracy loss, this choice of $\beta$ could be too small. Those settings imply that the noise added is larger and the distance between queries and threshold shrinks, so our method is more likely to terminate earlier than desired. Smaller $\beta$ values will then intensify this issue as the candidate values increase more slowly. For the clipping application, we generally think using a value of $\beta = 1.01$ would be a more stable choice. In fact, replicating our empirical testing with this value actually improves our results in Table 1. Furthermore, increasing $\beta$ will also reduce the number of queries and thus the computational cost. In general, we find that setting $\beta \in [1.01, 1.001]$ gives good performance with $\beta = 1.01$ as a default for computational efficiency. We leave to future work a more thorough analysis of this parameter to fully optimize setting it with relation to data size and privacy parameter. Additionally, all our methods and proofs only require an increasing sequence of candidate values and it's possible that other potential sequences would be even more effective. For example, if tight upper and lower bounds are known on the data, such as within the recursive computation of multiple quantiles, then it likely makes more sense to simply uniformly partition the interval and check in increasing order. But we leave more consideration upon this to future work as well.

Our choice of $q = 0.99$ in the differentially private sum experiments was also to maintain consistency with the choices for the previous method but also to keep variance of the estimates lower. This will create some negative bias as we then expect to clip the data. As we can see in our illustrative example Figure 3, the PDF will exponentially decrease once we pass the true quantile value, but will do so less sharply once all the queries have value $n$. Accordingly, setting $q = 1.0$ would add slightly more variance to the estimation but initial testing showed improvement in error. However, if the user would prefer slightly higher variance to avoid negative bias, then setting the threshold at $n$ or even $n + 1/\varepsilon$, would make it far more likely that the process terminates with a value slightly above the maximum. This is particularly useful for heavy-tailed data, where clipping at the 99th percentile can have an out-sized impact on the bias.

## Acknowledgments and Disclosure of Funding

We thank our colleagues Matthew Clegg, James Honaker, and Kenny Leftin for helpful discussions and feedback. We also thank anonymous reviewers for their helpful feedback.

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

# A Maximal Quantiles Estimation

In this section, we specifically consider applying our unbounded quantile estimation to computing the highest quantiles. This is commonly needed for clipping the dataset to privately compute the sum and mean, which we empirically test in Section 5.3. Our primary goal here is to build intuition upon why our approach fundamentally gives more accurate and robust estimations of the highest quantiles when only loose upper bounds on the data are known. We first give more details upon the previous methods, particularly the EMQ method. Then we give a more detailed look at how these methods are negatively affected by loose upper bounds and why ours performs well in comparison.

## A.1 Previous techniques

The most effective and practical previous method for quantile estimation, EMQ, builds a distribution over an assumed bounded range $[a, b]$ through an invocation of the exponential mechanism. Assuming the data is sorted, it will partition the range based upon the data and select an interval $[x_j, x_{j+1}]$ with probability proportional to

$$\exp\left(-\frac{\varepsilon|j - qn|}{2}\right)(x_{j+1} - x_j)$$

where the intervals $[a, x_1]$ and $[x_n, b]$ are also considered. A uniformly random point is then drawn from within the selected interval. Note that this utility function is not monotonic and will not enjoy the same improved privacy bounds.

There are other approaches that recursively partition the data range while computing statistics on each partition, then aggregate these statistics across logarithmic levels to reduce the noise for quantile computation Dwork et al. (2010); Chan et al. (2011). Another technique is to instead utilize the local sensitivity while maintaining privacy by using a smoothing procedure that can also be used for computing quantiles Nissim et al. (2007). There is also a similar approach to ours within Chen et al. (2016) for bounding an unbounded dataset that increases the bounds by a factor of 2 each iteration but we will discuss in Section A.2 why this variant performs substantially worse.

## A.2 Effect of bounds upon maximum quantiles

As mentioned, EMQ will construct a probability distribution over the range $[a, b]$. The corresponding PDF is a unimodal step function of the intervals defined by the data with the peak interval $[x_j, x_{j+1}]$ being such that $j$ closest to $qn$. Note then that if $b >> x_n$ then the probability of selecting $[x_n, b]$ increases dramatically. The exponential decay as the PDF moves away from the peak interval diminishes the impact of $[x_n, b]$ significantly if $n$ is far away from $qn$. However, for computing the highest quantiles, we want $q$ close to 1 by definition, and the looseness of the upper bound drastically effects the accuracy.

In contrast, as soon as the candidate value for our method exceeds the true quantile value, each successive query will have an output of at least $qn$ which is the threshold. The probability of continuing for $\lambda$ more queries then decreases in $\lambda$. [4] For ease of comparison, we can modify our Algorithm 2 such that in the last step, the output is drawn uniformly from $[\beta^{k-1}, \beta^k]$, assuming $\ell = 1$ for simplicity. This would then create a continuous probability distribution that would similarly be a step function with each interval $[\beta^{k-1}, \beta^k]$ being a step.

For better intuition we plot the approximate PDF of each in Figure 3. We use the Gumbel noise in the call to AboveThreshold to take advantage of the closed form expression in Lemma B.2 for easier PDF computation.

As we can see in Figure 3, the upper bound, $b$, increasing from 10 to 20 dramatically increases the probability of selecting a point in $[x_n, b]$ which changes the normalization for the other intervals, significantly altering the PDF of EMQ. Given that our method is unbounded, the PDF for UQE stays

---

[4]We expect $\lambda$ to most likely be small and this only adds a factor of $\beta^\lambda$ additional error. Note that setting $\beta$ too high, such as $\beta = 2$ which gives a similar algorithm to Chen et al. (2016), implies that even taking five additional queries will lead to a value that is over 32 times too large.

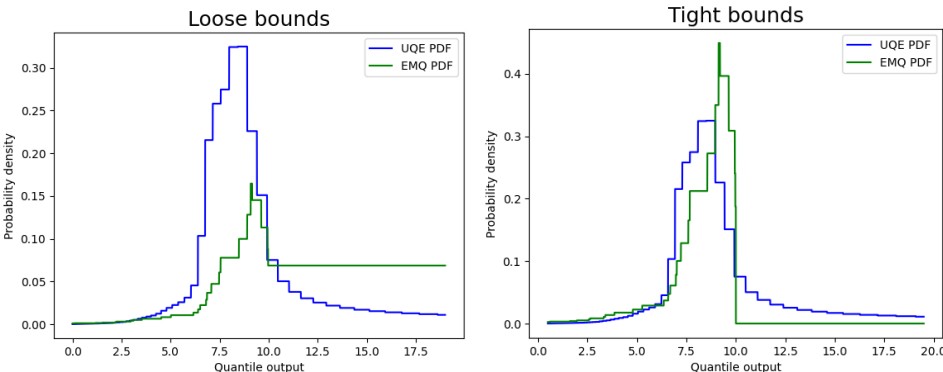

Figure 3: Illustrative example of how the approximate PDF of the previous method, EMQ is affected by looser upper bounds compared to our unbounded method UQE. A small amount of data was drawn uniformly from $[0, 10]$ and we set $q = 0.9$, so accurate output would be about 9. The left and right side assumed a range of $[0, 20]$ and $[0, 10]$ respectively for EMQ.

the same in both figures and sees the expected exponential decay once the candidate quantile passes the true quantile.

There are also alternative methods for bounding the data range by computing the interquartile range more accurately and scaling up. However these methods make strong assumptions upon the data distribution being close to the normal distribution as well as bounds upon the third moment Smith (2011). For real-world datasets, the tails of the data can vary significantly, and scaling up in a data-agnostic manner can often be similarly inaccurate. We also found this to be true in our experiments.

The smooth sensitivity framework will run into similar issues when $q$ is closer to 1 because fewer data points need to be changed to push the quantile value to the upper bound. If this upper bound is large, then the smooth local sensitivity will still be substantial. Aggregate tree methods are slightly more robust to loose upper bounds as they aren't partitioning specific to the data. However, for accurate estimates the partitioning of the range still requires the data be well-distributed across the evenly split partitions, which is significantly affected by loose upper bounds.

## B   Improved analysis for Sparse Vector Technique

In this section we complete the analysis for improving the privacy bounds for sparse vector technique with monotonic queries. We will first give a more generalized characterization of query properties by which we can improve the privacy bounds for AboveThreshold. While this characterization is more complex, we will immediately utilize it in Section C to improve the privacy bounds of our methods under an alternate common definition of neighboring. Additionally, the class of queries that we apply it to are not monotonic, so improvements can be made in a multitude of settings. Next we will review the definition of range-bounded, a property of privacy mechanisms that can be used to improve composition bounds, and apply it to our setting. Finally we will utilize these results to show that the privacy analysis of the sparse vector technique can be improved more generally, but also specifically for monotonic queries.

### B.1   Generalized characterization

We provide a more general characterization for the privacy loss of AboveThreshold that is specific to the input stream of queries. This will be achieved by providing a one-sided privacy parameter for each pair of neighboring datasets where order matters. For a given pair $x, x'$, our goal will be to upper bound the output distribution of $x$ by the output distribution of $x'$ up to an exponential factor of the one-sided privacy parameter. The specificity of the parameter to the neighboring datasets will reduce conciseness but it is for this reason that we will be able to further improve privacy analysis. This precise characterization will also help improve bounds for our method in Section C without requiring onerous analysis.

**Definition B.1.** *For a stream of queries $\{f_i\}$ with sensitivity $\Delta$, we define our one-sided privacy loss for neighboring data sets $x, x'$ as*

$$\varepsilon(x, x') = \max_k \left( \frac{\varepsilon_1}{\Delta} \Delta_k(x, x') + \frac{\varepsilon_2}{\Delta} \max\{0, \Delta_k(x, x') - (f_k(x') - f_k(x))\} \right)$$

*where $\Delta_k(x, x') = \max_{i<k} \max\{0, f_i(x') - f_i(x)\}$.*

Given that our definition is meant to encompass the one-sided privacy loss for AboveThreshold we will now prove that fact for Expo and Gumbel noise (and Lap follows equivalently to Expo). We first prove this conjecture for Expo noise.

**Lemma B.1.** *For a stream of queries $\{f_i\}$ with sensitivity $\Delta$, threshold $T$ and neighboring data sets $x, x'$, if we run Algorithm 1 with Expo noise, then for any given outcome $\{\perp^{k-1}, \top\}$ we have*

$$Pr\left[\text{AboveThreshold}(x, \{f_i\}, T) = \{\perp^{k-1}, \top\}\right] \leq$$
$$\exp(\varepsilon(x, x')) Pr\left[\text{AboveThreshold}(x', \{f_i\}, T) = \{\perp^{k-1}, \top\}\right]$$

*Proof.* Let $v_i \sim \text{Expo}(\Delta/\varepsilon_2)$ denote the noise drawn for query $f_i$, and let $v \sim \text{Expo}(\Delta/\varepsilon_1)$ denote the noise drawn for the threshold. Further let $v_{i\leq k}$ denote all $v_i$ such that $i \leq k$. By construction, we have

$$\Pr\left[\text{AboveThreshold}(x, \{f_i\}, T) = \{\perp^{k-1}, \top\}\right] = \Pr_{v, v_{i\leq k}}\left[\max_{i<k} f_i(x) + v_i < T + v < f_k(x) + v_k\right]$$

We will fix the randomness of $v_{i<k}$, denoting $v_i$ such that $i < k$, for datasets $x$ and $x'$, and let $\tau = \max_{i<k} f_i(x) + v_i$ and $\tau' = \max_{i<k} f_i(x') + v_i$. It suffices then to show that for any fixed randomness $v_{i<k}$ we have

$$\Pr_{v, v_k}\left[\tau < T + v < f_k(x) + v_k\right] \leq \exp(\varepsilon(x, x')) \Pr_{v, v_k}\left[\tau' < T + v < f_k(x') + v_k\right]$$

Let $\Delta_1 = \max\{0, \tau' - \tau\}$ and $\Delta_2 = \max\{0, \Delta_1 - (f_k(x') - f_k(x))\}$ and let $v' = v + \Delta_1$ and $v'_k = v_k + \Delta_2$. It suffices to show that for every pair of draws $v, v_k$ that satisfies $\tau < T + v < f_k(x) + v_k$, then the corresponding $v', v'_k$ are i) non-negative, ii) satisfy $\tau' < T + v' < f_k(x') + v'_k$ and iii)

$$p_{\text{Expo}}(v; \Delta/\varepsilon_1) \cdot p_{\text{Expo}}(v_k; \Delta/\varepsilon_2) \leq \exp(\varepsilon(x, x')) p_{\text{Expo}}(v'; \Delta/\varepsilon_1) \cdot p_{\text{Expo}}(v'_k; \Delta/\varepsilon_2)$$

We note that condition iii) does not require any scalar on the RHS as our change of variable is a shift for each so the Jacobian of the mapping is the identity matrix with determinant of 1.

Condition i) follows from the fact that $v, v_k$ are non-negative because they're drawn from the exponential distribution, and $\Delta_1, \Delta_2$ are non-negative by construction.

For condition ii), if $\tau < T + v$ we must have $\tau' < T + v + \Delta_1$ by definition of $\Delta_1$. Similarly, if $T + v < f_k(x) + v_k$ then $T + v + \Delta_1 < f_k(x') + v_k + \Delta_2$ because $\Delta_2 \geq \Delta_1 - (f_k(x') - f_k(x))$.

For condition iii), the PDF of the exponential distribution implies $p_{\text{Expo}}(v; \Delta/\varepsilon_1) \leq \exp(\frac{\varepsilon_1 \Delta_1}{\Delta}) p_{\text{Expo}}(v'; \Delta/\varepsilon_1)$ and $p_{\text{Expo}}(v_k; \Delta/\varepsilon_2) \leq \exp(\frac{\varepsilon_2 \Delta_2}{\Delta}) p_{\text{Expo}}(v'_k; \Delta/\varepsilon_2)$. Due to the fixing of randomness of $v_{i<k}$, we have that $\tau' - \tau \leq \max_{i<k}(f_i(x') - f_i(x))$ so $\Delta_1 \leq \Delta_k(x, x')$ from Definition B.1 and $\Delta_2 \leq \max\{0, \Delta_k(x, x') - (f_k(x') - f_k(x))\}$ which implies condition iii). $\square$

This proof then easily applies to Lap as well. The proof for Gumbel will differ though as we no longer have similar closeness of PDF properties, but it will follow from the fact that we can give a closed form expression for each probability.

**Lemma B.2.** *Given a dataset $x$, a stream of queries $\{f_i\}$ and threshold $T$, for any outcome $\{\perp^{k-1}, \top\}$ from running AboveThreshold with Gumbel noise and $\varepsilon = \varepsilon_1 = \varepsilon_2$, then*

$$Pr\left[\text{AboveThreshold}(x,\{f_i\},T) = \{\perp^{k-1},\top\}\right] =$$

$$\frac{\exp(\frac{\varepsilon}{\Delta}f_k(x))}{\exp(\frac{\varepsilon}{\Delta}T) + \sum_{i=1}^{k}\exp(\frac{\varepsilon}{\Delta}f_i(x))} \cdot \frac{\exp(\frac{\varepsilon}{\Delta}T)}{\exp(\frac{\varepsilon}{\Delta}T) + \sum_{i=1}^{k-1}\exp(\frac{\varepsilon}{\Delta}f_i(x))}$$

*Proof.* We add noise $\text{Gumbel}(\Delta/\varepsilon)$ to all $f_i(x)$ and the threshold $T$, and in order to output $\{\perp^{k-1},\top\}$ we must have that for the first $k$ queries, the noisy value of $f_k(x)$ is the largest and the noisy threshold is the second largest. It is folklore in the literature that adding $\text{Gumbel}$ noise (with the same noise parameter) and choosing the largest index is equivalent to the exponential mechanism. This can also be extended to showing that adding $\text{Gumbel}$ noise and choosing the top-k indices in order is equivalent to the peeling exponential mechanism. The peeling exponential mechanism first selects the top index using the exponential mechanism and removes it from the candidate set, repeating iteratively until accumulating the top-k indices. A formal proof of this folklore result is also provided in Lemma 4.2 of Durfee and Rogers (2019). Applying this result and the definition of the exponential mechanism gives the desired equality.

$\square$

We then utilize this closed form expression to get our desired one-sided privacy bounds when using Gumbel noise.

**Lemma B.3.** *For a stream of queries $\{f_i\}$ with sensitivity $\Delta$, threshold $T$ and neighboring data sets $x, x'$, if we run Algorithm 1 with $\text{Gumbel}$ noise with $\varepsilon_1 = \varepsilon_2$, then for any outcome $\{\perp^{k-1},\top\}$ we have*

$$Pr\left[\text{AboveThreshold}(x,\{f_i\},T) = \{\perp^{k-1},\top\}\right] \leq$$

$$\exp(\varepsilon(x,x'))Pr\left[\text{AboveThreshold}(x',\{f_i\},T) = \{\perp^{k-1},\top\}\right]$$

*Proof.* Let $\varepsilon = \varepsilon_1 = \varepsilon_2$. From Lemma B.2 we know that

$$Pr\left[\text{AboveThreshold}(x,\{f_i\},T) = \{\perp^{k-1},\top\}\right] =$$

$$\frac{\exp(\frac{\varepsilon}{\Delta}f_k(x))}{\exp(\frac{\varepsilon}{\Delta}T) + \sum_{i=1}^{k}\exp(\frac{\varepsilon}{\Delta}f_i(x))} \cdot \frac{\exp(\frac{\varepsilon}{\Delta}T)}{\exp(\frac{\varepsilon}{\Delta}T) + \sum_{i=1}^{k-1}\exp(\frac{\varepsilon}{\Delta}f_i(x))}$$

Similar to the proof of Lemma B.1, we let $\Delta_1 = \max\{0, \max_{i<k}(f_i(x') - f_i(x))\}$ and $\Delta_2 = \max\{0, \Delta_1 + (f_k(x) - f_k(x'))\}$. We first want to show that

$$\frac{\exp(\frac{\varepsilon}{\Delta}T)}{\exp(\frac{\varepsilon}{\Delta}T) + \sum_{i=1}^{k-1}\exp(\frac{\varepsilon}{\Delta}f_i(x))} \leq e^{\frac{\varepsilon\Delta_1}{\Delta}}\frac{\exp(\frac{\varepsilon}{\Delta}T)}{\exp(\frac{\varepsilon}{\Delta}T) + \sum_{i=1}^{k-1}\exp(\frac{\varepsilon}{\Delta}f_i(x'))} \tag{1}$$

This inequality reduces to showing $\exp(\frac{\varepsilon}{\Delta}T) + \sum_{i=1}^{k-1}\exp(\frac{\varepsilon}{\Delta}f_i(x')) \leq e^{\frac{\varepsilon\Delta_1}{\Delta}}(\exp(\frac{\varepsilon}{\Delta}T) + \sum_{i=1}^{k-1}\exp(\frac{\varepsilon}{\Delta}f_i(x)))$. This holds from the fact that $\exp(\frac{\varepsilon}{\Delta}T) \leq \exp(\varepsilon\Delta_1/\Delta)\exp(\frac{\varepsilon}{\Delta}T)$ because $\varepsilon\Delta_1/\Delta \geq 0$ and $\exp(\frac{\varepsilon}{\Delta}f_i(x')) \leq \exp(\varepsilon\Delta_1/\Delta)\exp(\frac{\varepsilon}{\Delta}f_i(x))$ for all $i < k$ by our definition of $\Delta_1$.

$$\frac{\exp(\frac{\varepsilon}{\Delta}f_k(x))}{\exp(\frac{\varepsilon}{\Delta}T) + \sum_{i=1}^{k}\exp(\frac{\varepsilon}{\Delta}f_i(x))} \leq e^{\frac{\varepsilon\Delta_2}{\Delta}}\frac{\exp(\frac{\varepsilon}{\Delta}f_k(x'))}{\exp(\frac{\varepsilon}{\Delta}T) + \sum_{i=1}^{k}\exp(\frac{\varepsilon}{\Delta}f_i(x'))} \tag{2}$$

First we let $\Delta_1' = \max\{0, \max_{\ell \leq k}(f_\ell(x') - f_\ell(x))\}$ and $\Delta_2' = f_k(x) - f_k(x')$, and we will achieve our desired inequality (2) by bounding the numerator with respect to $\Delta_2'$ and the denominator with respect to $\Delta_1'$. We have $\exp(\frac{\varepsilon}{\Delta}f_k(x)) = \exp(\varepsilon\Delta_2'/\Delta)\exp(\frac{\varepsilon}{\Delta}f_k(x'))$ by definition of $\Delta_2'$. Further $\exp(\frac{\varepsilon}{\Delta}f_i(x')) \leq \exp(\varepsilon\Delta_1'/\Delta)\exp(\frac{\varepsilon}{\Delta}f_i(x))$ by definition of $\Delta_1'$, so

$$\exp(\frac{\varepsilon}{\Delta}T) + \sum_{i=1}^{k}\exp(\frac{\varepsilon}{\Delta}f_i(x')) \leq e^{\frac{\varepsilon\Delta_1'}{\Delta}}(\exp(\frac{\varepsilon}{\Delta}T) + \sum_{i=1}^{k}\exp(\frac{\varepsilon}{\Delta}f_i(x)))$$

because $\epsilon \Delta_1'/\Delta \geq 0$. In order to show our desired inequality (2) it then suffices to show $\Delta_2 \geq \Delta_1' + \Delta_2'$. By definition we know $\Delta_2 \geq \Delta_1 + \Delta_2'$. Furthermore, if $\Delta_1' > \Delta_1$ then $\Delta_1' = f_k(x') - f_k(x) = -\Delta_2'$ which implies $\Delta_1' + \Delta_2' = 0$, and we know $\Delta_2 \geq 0$ by construction.

Combining inequality (1) and (2) along with the fact that $\Delta_1 = \Delta_k(x, x')$ from Definition B.1 completes the proof.

$\square$

Now that we've bounded the one-sided privacy loss for neighboring datasets for each type of noise, this will immediately imply an overall privacy bound over all neighbors that utilizes this general characterization.

**Corollary B.1.** *For a stream of queries* $\{f_i\}$ *with sensitivity* $\Delta$ *and threshold* $T$, *Algorithm* 1 *with* Lap, Gumbel, *or* Expo *noise is* $\varepsilon$-*DP where*

$$\varepsilon = \max_{x \sim x'} \varepsilon(x, x')$$

*and for* Gumbel *noise we must have* $\varepsilon_1 = \varepsilon_2$.

## B.2 Range-bounded definition and properties

The definition of *range-bounded* was originally introduced in Durfee and Rogers (2019) to tighten the privacy bounds on the composition of exponential mechanisms. The analysis was further extended in Dong et al. (2020) to give the optimal composition of range-bounded mechanisms. This definition was then unified with other definitions in Cesar and Rogers (2021).

**Definition B.2** (Durfee and Rogers (2019)). *A mechanism* $M : \mathcal{X} \to \mathcal{Y}$ *is* $\varepsilon$-*range-bounded if for any neighboring datasets* $x, x' \in \mathcal{X}$ *and outcomes* $y, y' \in \mathcal{Y}$:

$$\frac{Pr\left[M(x) = y\right]}{Pr\left[M(x') = y\right]} \leq e^\varepsilon \frac{Pr\left[M(x) = y'\right]}{Pr\left[M(x') = y'\right]}$$

If we have bounds upon this property that are stronger than those immediately implied by DP, then it was shown in Dong et al. (2020) that substantial improvements can be made in bounding the overall DP over the composition of these mechanisms. While these improvements can be applied to our results here as well, we focus upon the privacy bounds with respect to zCDP as it is much cleaner to work with in providing strong composition bounds. From the proof of Lemma 3.4 in Cesar and Rogers (2021) we see that the range-bounded property can give a significant improvement in zCDP properties compared to similar DP guarantees.

**Proposition 2** (Cesar and Rogers (2021)). *If a mechanism* $M$ *is* $\varepsilon$-*range-bounded then it is* $\frac{1}{8}\varepsilon^2$-*zCDP*.

Our general characterization is already set up to provide range-bounded properties of AboveThreshold. This will allow us to give tighter guarantees on the range-boundedness which can be translated to better privacy composition bounds.

**Corollary B.2.** *For a stream of queries* $\{f_i\}$ *and threshold* $T$, *Algorithm* 1 *with* Lap, Gumbel, *or* Expo *noise is* $\varepsilon$-*range-bounded where*

$$\varepsilon = \max_{x \sim x'} \left(\varepsilon(x, x') + \varepsilon(x', x)\right)$$

*and for* Gumbel *noise we must have* $\varepsilon_1 = \varepsilon_2$.

*Proof.* We can rewrite the inequality from Definition B.2 as equivalently

$$\Pr\left[M(x) = y\right] \cdot \Pr\left[M(x') = y'\right] \leq e^\varepsilon \Pr\left[M(x') = y\right] \cdot \Pr\left[M(x) = y'\right]$$

Our claim then follows immediately by applying Lemma B.1 and Lemma B.3 $\square$

## B.3 Improved composition analysis of SVT

In this section we complete our analysis for improving the privacy bounds for sparse vector technique. While our generalized characterization was in a more complex form, we simplify it here for general queries and monotonic queries. This will then match up to the privacy guarantees that we gave in Section 3 for AboveThreshold.

**Lemma B.4.** *Given a stream of queries $\{f_i\}$ with sensitivity $\Delta$ then for any neighboring datasets $\varepsilon(x, x') \leq \varepsilon_1 + 2\varepsilon_2$. If the queries are monotonic then for any neighboring datasets $\varepsilon(x, x') \leq \varepsilon_1 + \varepsilon_2$, and furthermore, $\varepsilon(x, x') + \varepsilon(x', x) \leq \varepsilon_1 + 2\varepsilon_2$.*

*Proof.* By Definition 2.2 we must always have $\Delta_k(x, x') \leq \Delta$ for any neighboring datasets. Furthermore, we must also have $f_k(x') - f_k(x) \geq -\Delta$ for any neighboring datasets. This implies $\varepsilon(x, x') \leq \varepsilon_1 + 2\varepsilon_2$ for any neighboring datasets.

Now assume the queries are monotonic. Without loss of generality assume $f_k(x) \geq f_k(x')$ for all $k$. Therefore $\Delta_k(x, x') = 0$ and $f_k(x') - f_k(x) \geq -\Delta$, which implies $\varepsilon(x, x') \leq \varepsilon_2$. Similarly we have $\Delta_k(x', x) \leq \Delta$ but also $f_k(x) - f_k(x') \geq 0$, which implies $\varepsilon(x', x) \leq \varepsilon_1 + \varepsilon_2$. Combining these implies $\varepsilon(x', x) \leq \varepsilon_1 + \varepsilon_2$ and $\varepsilon(x', x) + \varepsilon(x, x') \leq \varepsilon_1 + 2\varepsilon_2$ for any neighboring datasets. $\square$

This lemma along with Corollary B.1 immediately imply Corollary 3.1 and Lemma 3.2

*Proof of Theorem 1.* From Lemma B.4 and Corollary B.2 we have that Algorithm 1 is $\varepsilon_1 + 2\varepsilon_2$-range-bounded. Applying Proposition 2 then gives the zCDP guarantees. $\square$

The sparse vector technique is simply an iterative call to AboveThreshold and as such the privacy bounds will come from the composition. By analyzing the composition through zCDP which has become commonplace in the privacy community, we can immediately improve the privacy guarantees for monotonic queries in the sparse vector technique. Furthermore we can also improve the privacy guarantees under the standard definition using the improved composition bounds for range-bounded mechanisms from Dong et al. (2020).

We also note here that the generalized Sparse Vector Technique given in Lyu et al. (2017) only adds noise once to the threshold and only has to scale the noise to the number of calls to AboveThreshold for the queries and not the threshold. Our analysis can extend to give the same properties for Lap and Expo noise, but utilizing advanced composition properties will give far more accuracy. More specifically, the threshold and queries having noise proportional to $\sqrt{c}/\varepsilon$, is preferable to all the queries having noise proportional to $c/\varepsilon$, and only the threshold proportional to $1/\varepsilon$, where $c$ is the number of calls to AboveThreshold.

## B.4 Iterative exponential mechanism

It is folklore that the peeling exponential mechanism is equivalent to taking the top-$k$ after adding Gumbel noise, as formally shown in Lemma 4.2 of Durfee and Rogers (2019). We show a similar property here for AboveThreshold with Gumbel noise that it is equivalent to iteratively running the exponential mechanism.

---

**Algorithm 3** Iterative Exponential Mechanism

---

**Require:** Input dataset $x$, a stream of queries $\{f_i : \mathcal{X} \to \mathbb{R}\}$ with sensitivity $\Delta$, and a threshold $T$
  1: **for** each query $i$ **do**
  2:     Run exponential mechanism with $\mathcal{Y} = \{0, ..., i\}$ where $q(x, 0) = T$ and $q(x, j) = f_j(x)$ for all $j > 0$
  3:     **if** $i$ is selected **then**
  4:         Output $\top$ and halt
  5:     **else**
  6:         Output $\bot$
  7:     **end if**
  8: **end for**

---

**Lemma B.5.** *If $\varepsilon_1 = \varepsilon_2 = \varepsilon/2$ then Algorithm 1 with* `Gumbel` *noise gives the equivalent output distribution to Algorithm 3.*

*Proof.* We first define

$$p_k = \frac{\exp(\frac{\varepsilon}{2\Delta} f_k(x))}{\exp(\frac{\varepsilon}{2\Delta} T) + \sum_{i=1}^{k} \exp(\frac{\varepsilon}{2\Delta} f_i(x))}$$

By construction, the probability of Algorithm 3 outputting $\{\perp^{k-1}, \top\}$ is equal to $p_k \prod_{i=1}^{k-1}(1 - p_i)$. Furthermore, by our definition of $p_k$ we have

$$1 - p_k = \frac{\exp(\frac{\varepsilon}{2\Delta} T) + \sum_{i=1}^{k-1} \exp(\frac{\varepsilon}{2\Delta} f_i(x))}{\exp(\frac{\varepsilon}{2\Delta} T) + \sum_{i=1}^{k} \exp(\frac{\varepsilon}{2\Delta} f_i(x))}$$

Through telescoping cancellation

$$\prod_{i=1}^{k-1}(1 - p_i) = \frac{\exp(\frac{\varepsilon}{2\Delta} T)}{\exp(\frac{\varepsilon}{2\Delta} T) + \sum_{i=1}^{k-1} \exp(\frac{\varepsilon}{2\Delta} f_i(x))}$$

From Lemma B.2 we then see that the output probabilities are equivalent.

$\square$

# C  Additional unbounded quantile estimation results

In this section we first extend our method to the fully unbounded setting by simply making two calls to `AboveThreshold` and essentially searching through both positive and negative numbers in each respective call. Next we show how our methods can also extend to the *add-subtract* definition of neighboring. Further, we'll apply our approach to the framework set up in Kaplan et al. (2022) for computing multiple quantiles and extend it to the *swap* definition.

## C.1  Fully unbounded quantile estimation

Recall that our unbounded quantile estimation algorithm assumed that the data was lower bounded. It then slowly increased that bound by a small percentage until the appropriate amount of data fell below the threshold for the given quantile. In order to extend to the fully unbounded setting, we will simply first apply this guess and check method to the positive numbers and then apply it to the negative numbers.

---
**Algorithm 4** Fully unbounded quantile mechanism
---
**Require:** Input dataset $x$, a quantile $q$, and parameter $\beta > 1$
 1: Run `AboveThreshold` with $x$, $T = qn$ and $f_i(x) = |\{x_j \in x | x_j + 1 < \beta^i\}|$
 2: Run `AboveThreshold` with $x$, $T = (1-q)n$ and $f_i(x) = |\{x_j \in x | x_j - 1 > -\beta^i\}|$
 3: If the first `AboveThreshold` halts at $k > 0$ then output $\beta^k - 1$
 4: If the second `AboveThreshold` halts at $k > 0$ then output $-\beta^k + 1$
 5: Otherwise return 0
---

Note that the second call could equivalently be achieved by flipping the sign of all the datapoints and again applying queries $f_i(x) = |\{x_j \in x | x_j + 1 < \beta^i\}|$. Therefore all our privacy guarantees from Section 4 will still apply, but composing over two calls to `AboveThreshold`, which is the Sparse Vector Technique.

In the first call to `AboveThreshold` we are assuming a lower bound of 0, and searching the positive numbers. If it terminates immediately then it is likely that more than a $q$th fraction of the data is below 0. We then symmetrically search through the negative numbers by assuming an upper bound of 0. If this halts immediately then it is likely that the quantile is already near 0. We could also apply other variants of this, such as three total calls to get the maximum and minimum of the data if we wanted full data bounds.

Further note that computing the smallest quantiles will be challenging for our algorithm because it may take many queries to get to the appropriate threshold, but each query will have a reasonable chance of terminating if $q$ is very small. To account for these queries in the lower-bounded setting, we can invert all the datapoints and instead search for quantile $1 - q$ on this transformed data, then invert our resulting estimate. We would also need to reduce our parameter $\beta$ a reasonable amount in this setting.

## C.2   Extension to *add-subtract* neighbors

We also extend our results to the *add-subtract* definition of neighboring datasets.

**Definition C.1.** *Datasets $x, x' \in \mathcal{X}$ are neighboring if one of them can be obtained from the other by adding or removing one individual's data.*

Under this definition we see that our threshold $T = qn$ is no longer fixed, so we will instead set each query to be $f_i(x) = |\{x_j \in x | x_j - \ell + 1 < \beta^i\}| - qn$. Unfortunately this query will no longer be monotonic, so we will instead take advantage of our more general characterization in Section B.1 to further tighten the privacy bounds in this setting.

**Lemma C.1.** *Given the stream of queries $f_i(x) = |\{x_j \in x | x_j - \ell + 1 < \beta^i\}| - qn$ with sensitivity $\Delta = 1$, for any neighboring datasets $x, x'$ under Definition C.1 and quantile $q \in [0, 1]$ we have $\varepsilon(x, x') \leq \max\{(1 - q)\varepsilon_1, q\varepsilon_1 + \varepsilon_2\}$.*

*Proof.* Without loss of generality, assume that $x$ has one more individuals data, so $x \in \mathbb{R}^n$ and $x' \in \mathbb{R}^{n-1}$. Let $g_i(x) = |\{x_j \in x | x_j - \ell + 1 < \beta^i\}|$. By construction we must have $g_i(x) - g_i(x') \in \{0, 1\}$ because $x$ has an additional datapoint. Furthermore, because the thresholds are increasing, if $g_k(x) - g_k(x') = 1$ then $g_i(x) - g_i(x') = 1$ for all $i > k$. Similarly if $g_k(x) - g_k(x') = 0$ then $g_i(x) - g_i(x') = 0$ for all $i < k$. We further see that $f_i(x) - f_i(x') = g_i(x) - g_i(x') - q$ for all $i$.

First consider the case when $g_k(x) - g_k(x') = 0$. We therefore have $g_i(x) - g_i(x') = 0$ for all $i < k$ so $\Delta_k(x, x') = q$, and also $\Delta_k(x, x') - (f_k(x') - f_k(x)) = 0$, so $\varepsilon(x, x') \leq q\varepsilon_1$. Similarly, we have $\Delta_k(x', x) = 0$ and $\Delta_k(x, x') - (f_k(x') - f_k(x)) = q$, so $\varepsilon(x', x) \leq q\varepsilon_2$.

Next consider the case when $g_k(x) - g_k(x') = 1$. Therefore we have $\Delta_k(x, x') \leq q$ and also $f_k(x') - f_k(x) = q - 1$ which implies $\Delta_k(x, x') - (f_k(x') - f_k(x)) \leq 1$, so $\varepsilon(x, x') \leq q\varepsilon_1 + \varepsilon_2$. Similarly, we have that $\Delta_k(x', x) \leq 1 - q$, and thus $\Delta_k(x', x) - (f_k(x) - f_k(x')) = 0$, so $\varepsilon(x', x) \leq (1 - q)\varepsilon_1$.

Combining these bounds gives our desired result.

$\square$

With these improved bounds we can then show that our methods also extend to the *add-subtract* definition of neighboring with tighter privacy guarantees.

**Corollary C.1.** *If we run Algorithm 2 with $\varepsilon_1 = \varepsilon_2$ for a quantile $q$ under Definition C.1 then it is $(q\varepsilon_1 + \varepsilon_2)$-DP.*

*Proof.* Combining Lemma C.1 and Corollary B.1 immediately imply the desired result

$\square$

## C.3   Extension to multiple quantile estimation

As previously established, the framework in Kaplan et al. (2022) is agnostic to the single quantile estimation method. However their proof is for the *add-subtract* neighbors, although they note that it can be extended easily to the *swap* neighbors. We also discuss here how it can be extended for completeness.

For the *swap* neighboring definition, at each level of the recursive partitioning scheme either two partitions differ under the *add-subtract* definition, or one partition differs under the *swap* definition. Applying their framework to our algorithm, we will instead compute all the thresholds in advance of the splitting. Accordingly these thresholds will remain fixed. If the thresholds are fixed then we can actually further improve our privacy bounds for the *add-subtract* definition. Once again we will use our more general characterization from Section B.1.

**Lemma C.2.** *Given the stream of queries $f_i(x) = |\{x_j \in x | x_j - \ell + 1 < \beta^i\}|$ with sensitivity $\Delta = 1$, for any neighboring datasets $x, x'$ under Definition C.1 we have $\varepsilon(x, x') \leq \max\{\varepsilon_1, \varepsilon_2\}$*

*Proof.* Without loss of generality, assume that $x$ has one more individuals data, so $x \in \mathbb{R}^n$ and $x' \in \mathbb{R}^{n-1}$. By construction we must have $f_i(x) - f_i(x') \in \{0, 1\}$ because $x$ has an additional datapoint. Furthermore, because the thresholds are increasing, if $f_k(x) - f_k(x') = 0$ then $f_i(x) - f_i(x') = 0$ for all $i < k$. Thus the case of $f_k(x) - f_k(x') = 0$ is easy.

Instead consider $f_k(x) - f_k(x') = 1$. We know $\Delta_k(x, x') = 0$ so $\Delta_k(x, x') - (f_k(x') - f_k(x)) = 1$, so $\varepsilon(x, x') \leq \varepsilon_2$. Further we must have $\Delta_k(x', x) \leq 1$ and $\Delta_k(x', x) - (f_k(x) - f_k(x')) = 0$, so $\varepsilon(x', x) \leq \varepsilon_1$. Combining these implies $\varepsilon(x', x) \leq \max\{\varepsilon_1, \varepsilon_2\}$ for any neighboring datasets.

$\square$

Applying these bounds we see that applying the framework of Kaplan et al. (2022) to the swap definition either leads to one composition of $(\varepsilon_1 + \varepsilon_2)$-DP for the *swap* definition or two compositions of $\max\{\varepsilon_1, \varepsilon_2\}$ for the *add-subtract* definition at each level. Setting $\varepsilon_1 = \varepsilon_2$ we then have that the privacy cost of computing $m$ quantiles with this framework will be equivalent to $\log(m + 1)$ compositions of $(\varepsilon_1 + \varepsilon_2)$-DP.

# D  Implementation code

For ease of implementation, we provide some simple python code to run our method with access to the respective `Noise` generator of the users choosing. In our experiments we used exponential noise from the numpy library.

```python
def unboundedQuantile(data, l, b, q, eps_1, eps_2):
  d = defaultdict(int)
  for x in data:
    i = math.log(x-l+1,b) // 1
    d[i] += 1

  t = q * len(data) + noise(1/eps_1)
  cur, i = 0, 0
  while True:
    cur += d[i]
    i += 1
    if cur + noise(1/eps_2) > t:
      break
  return b**i - l + 1
```

