# Unbounded Differentially Private Quantile and Maximum Estimation

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

[\texttt{AboveThreshold}(x, \{f_i\}, T)\right] = \{\perp^{k-1}, \top\} \leq \exp(\varepsilon(x, x')) Pr\left[\texttt{AboveThreshold}(x', \{f_i\}, T)\right] = \{\perp^{k-1}, \top\}$$

554 *Proof.* Let $v_i \sim \texttt{Expo}(\Delta/\varepsilon_2)$ denote the noise drawn for query $f_i$, and let $v \sim \texttt{Expo}(\Delta/\varepsilon_1)$ denote the
555 noise drawn for the threshold. We will fix the randomness of $v_{i<k}$ for datasets $x$ and $x'$, and let
556 $\tau = \max_{i<k} f_i(x) + v_i$ and $\tau' = \max_{i<k} f_i(x') + v_i$.

557 It suffices then to show that for any fixed randomness $v_{i<k}$ we have

$$\Pr_{v, v_k} \left[\tau < T + v < f_k(x) + v_k\right] \leq \exp(\varepsilon(x, x')) \Pr_{v, v_k} \left[\tau' < T + v < f_k(x') + v_k'\right]$$

558 We can then prove this by showing that for every pair of draws $v, v_k$ that satisfies $\tau < T + v < f_k(x) + v_k$,
559 there is a unique pair $v', v_k'$ that satisfies $\tau' < T + v' < f_k(x') + v_k'$ and that

$$p_{\texttt{Expo}}(v; \Delta/\varepsilon_1) \cdot p_{\texttt{Expo}}(v_k; \Delta/\varepsilon_2) \leq \exp(\varepsilon(x, x')) p_{\texttt{Expo}}(v'; \Delta/\varepsilon_1) \cdot p_{\texttt{Expo}}(v_k'; \Delta/\varepsilon_2)$$

560 By definition, draws from the exponential distribution must take non-negative values so we must
561 also ensure that $v' \geq v$ and $v_k' \geq v_k$. As such, we let $\Delta_1 = \max\{0, \tau' - \tau\}$ and $\Delta_2 = \max\{0, \Delta_1 - (f_k(x') -$

562    $f_k(x))$}, and our injective mapping will be $v' = v + \Delta_1$ and $v'_k = v_k + \Delta_2$. It is then straightforward to
563    see that if $\tau < T + v$ we must have $\tau' < T + v + \Delta_1$. Similarly, if $T + v < f_k(x) + v_k$ then $T + v + \Delta_1 <$
564    $f_k(x') + v_k + \Delta_2$ because $\Delta_2 \geq \Delta_1 - (f_k(x') - f_k(x))$. The PDF of the exponential distribution then
565    gives us $p_{\mathsf{Expo}}(v; \Delta/\varepsilon_1) \leq \exp(\frac{\varepsilon_1 \Delta_1}{\Delta}) p_{\mathsf{Expo}}(v'; \Delta/\varepsilon_1)$ and $p_{\mathsf{Expo}}(v_k; \Delta/\varepsilon_2) \leq \exp(\frac{\varepsilon_2 \Delta_2}{\Delta}) p_{\mathsf{Expo}}(v'_k; \Delta/\varepsilon_2)$

566    Due to the fixing of randomness, we have that $\tau' - \tau \leq \max_{i<k}(f_i(x') - f_i(x))$ so $\Delta_1 \leq \Delta_k(x, x')$,
567    which implies our desired inequality above.

568    $\square$

569    This proof then easily applies to $\mathsf{Lap}$ as well. The proof for $\mathsf{Gumbel}$ will differ though as we no
570    longer have similar closeness of PDF properties, but can instead utilize our closed form expressions.

571    **Lemma B.2.** *Given a dataset $x$, a stream of queries $\{f_i\}$ and threshold $T$, for any outcome $\{\perp^{k-1}, \top\}$*
572    *from running* $\mathsf{AboveThreshold}$ *with* $\mathsf{Gumbel}$ *noise and $\varepsilon = \varepsilon_1 = \varepsilon_2$, then*

$$Pr\left[\mathsf{AboveThreshold}(x, \{f_i\}, T) = \{\perp^{k-1}, \top\}\right] =$$
$$\frac{\exp(\frac{\varepsilon}{\Delta} f_k(x))}{\exp(\frac{\varepsilon}{\Delta} T) + \sum_{i=1}^{k} \exp(\frac{\varepsilon}{\Delta} f_i(x))} \cdot \frac{\exp(\frac{\varepsilon}{\Delta} T)}{\exp(\frac{\varepsilon}{\Delta} T) + \sum_{i=1}^{k-1} \exp(\frac{\varepsilon}{\Delta} f_i(x))}$$

573    *Proof.* We add noise $\mathsf{Gumbel}(\Delta/\varepsilon)$ to all $f_i(x)$ and the threshold $T$, and in order to output $\{\perp^{k-1}, \top\}$
574    we must have that for the first $k$ queries, the noisy value of $f_k(x)$ is the largest and the noisy
575    threshold is the second largest. It is folklore in the literature that adding $\mathsf{Gumbel}$ noise (with the
576    same noise parameter) and choosing the largest index is equivalent to the exponential mechanism.
577    This can also be extended to showing that adding $\mathsf{Gumbel}$ noise and choosing the top-$k$ indices in
578    order is equivalent to the peeling exponential mechanism. The peeling exponential mechanism
579    first selects the top index using the exponential mechanism and removes it from the candidate set,
580    repeating iteratively until accumulating the top-$k$ indices. A formal proof of this folklore result is
581    also provided in Lemma 4.2 of Durfee and Rogers (2019). Applying this result and the definition of
582    the exponential mechanism gives the desired equality.

583    $\square$

584    Interestingly, we can similarly show that $\mathsf{AboveThreshold}$ with $\mathsf{Gumbel}$ noise is equivalent to an
585    iterative exponential mechanism that we provide in Section B.4. The closed form expression will
586    actually allow for a slightly improved characterization that we also utilize in Section C.2.

587    **Lemma B.3.** *For a stream of queries $\{f_i\}$ with sensitivity $\Delta$, threshold $T$ and neighboring data sets*
588    $x, x'$, *if we run Algorithm 1 with* $\mathsf{Gumbel}$ *noise with $\varepsilon_1 = \varepsilon_2$, then for any outcome $\{\perp^{k-1}, \top\}$ we have*

$$Pr[\mathsf{AboveThreshold}(x, \{f_i\}, T)] = \{\perp^{k-1}, \top\} \leq \exp(\varepsilon(x, x')) Pr\left[\mathsf{AboveThreshold}(x', \{f_i\}, T)\right] = \{\perp^{k-1}, \top\})$$

589    *and further we can relax $\Delta_k(x, x')$ to $\max_{i<k}(f_i(x') - f_i(x))$.*

590    *Proof.* Let $\varepsilon = \varepsilon_1 = \varepsilon_2$. From Lemma B.2 we know that

$$\Pr\left[\mathsf{AboveThreshold}(x, \{f_i\}, T) = \{\perp^{k-1}, \top\}\right] =$$
$$\frac{\exp(\frac{\varepsilon}{\Delta} f_k(x))}{\exp(\frac{\varepsilon}{\Delta} T) + \sum_{i=1}^{k} \exp(\frac{\varepsilon}{\Delta} f_i(x))} \cdot \frac{\exp(\frac{\varepsilon}{\Delta} T)}{\exp(\frac{\varepsilon}{\Delta} T) + \sum_{i=1}^{k-1} \exp(\frac{\varepsilon}{\Delta} f_i(x))}$$

591    Similar to the proof of Lemma B.1, we let $\Delta_1 = \max_{i<k}(f_i(x') - f_i(x))$ and $\Delta_2 = \max\{0, \Delta_1 + (f_k(x) -$
592    $f_k(x'))\}$. We first want to show that

$$\frac{\exp(\frac{\varepsilon}{\Delta} T)}{\exp(\frac{\varepsilon}{\Delta} T) + \sum_{i=1}^{k-1} \exp(\frac{\varepsilon}{\Delta} f_i(x))} \leq e^{\frac{\varepsilon \Delta_1}{\Delta}} \frac{\exp(\frac{\varepsilon}{\Delta} T)}{\exp(\frac{\varepsilon}{\Delta} T) + \sum_{i=1}^{k-1} \exp(\frac{\varepsilon}{\Delta} f_i(x'))}$$

This inequality reduces to $\sum_{i=1}^{k-1} \exp(\frac{\varepsilon}{\Delta} f_i(x')) \leq \exp(\varepsilon\Delta_1/\Delta) \sum_{i=1}^{k-1} \exp(\frac{\varepsilon}{\Delta} f_i(x))$ which follows from our definition of $\Delta_1$. Next we want to show that

$$\frac{\exp(\frac{\varepsilon}{\Delta} f_k(x))}{\exp(\frac{\varepsilon}{\Delta} T) + \sum_{i=1}^{k} \exp(\frac{\varepsilon}{\Delta} f_i(x))} \leq e^{\frac{\varepsilon\Delta_2}{\Delta}} \frac{\exp(\frac{\varepsilon}{\Delta} f_k(x'))}{\exp(\frac{\varepsilon}{\Delta} T) + \sum_{i=1}^{k} \exp(\frac{\varepsilon}{\Delta} f_i(x'))}$$

First we let $\Delta_1' = \max_{\ell \leq k}(f_\ell(x') - f_\ell(x))$ and $\Delta_2' = f_k(x) - f_k(x')$. We see that we equivalently have $\Delta_2 = \Delta_1' + \Delta_2'$ because if $\Delta_1' > \Delta_1$ then $\Delta_1' = f_k(x') - f_k(x) = -\Delta_2'$, and $\Delta_1 - \Delta_1' < 0$ which implies $\Delta_1' + \Delta_2' = \max\{0, \Delta_1 + \Delta_2'\} = 0$. Furthermore, we see that $\exp(\frac{\varepsilon}{\Delta} f_k(x)) = \exp(\varepsilon\Delta_2'/\Delta) \exp(\frac{\varepsilon}{\Delta} f_k(x'))$ and $\sum_{i=1}^{k} \exp(\frac{\varepsilon}{\Delta} f_i(x')) \leq \exp(\varepsilon\Delta_1'/\Delta) \sum_{i=1}^{k} \exp(\frac{\varepsilon}{\Delta} f_i(x))$ which implies the inequality above.

Combining our two inequalities along with the fact that $\Delta_1 \leq \Delta_k(x, x')$ completes the proof.

$\square$

Now that we've bounded the one-sided privacy loss for neighboring datasets for each type of noise, this will immediately imply an overall privacy bound over all neighbors that utilizes this general characterization.

**Corollary B.1.** *For a stream of queries $\{f_i\}$ with sensitivity $\Delta$ and threshold $T$, Algorithm 1 with* Lap, Gumbel, *or* Expo *noise is $\varepsilon$-DP where*

$$\varepsilon = \max_{x \sim x'} \varepsilon(x, x')$$

*and for* Gumbel *noise we must have $\varepsilon_1 = \varepsilon_2$.*

## B.2  Range-bounded definition and properties

The definition of *range-bounded* was originally introduced in Durfee and Rogers (2019) to tighten the privacy bounds on the composition of exponential mechanisms. The analysis was further extended in Dong et al. (2020) to give the optimal composition of range-bounded mechanisms. This definition was then unified with other definitions in Cesar and Rogers (2021).

**Definition B.2** (Durfee and Rogers (2019)). *A mechanism $M : \mathcal{X} \to \mathcal{Y}$ is $\varepsilon$-range-bounded if for any neighboring datasets $x, x' \in \mathcal{X}$ and outcomes $y, y' \in \mathcal{Y}$:*

$$\frac{Pr[M(x) = y]}{Pr[M(x') = y]} \leq e^\varepsilon \frac{Pr[M(x) = y']}{Pr[M(x') = y']}$$

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

') = 0$. Also note the relaxation for Gumbel noise in Lemma B.3. We then have $\Delta_k(x', x) = -q$ and thus $\Delta_k(x, x') - (f_k(x') - f_k(x)) = 0$, which instead implies $\varepsilon(x', x) \leq 0$ for this case. Therefore under these conditions, for neighbors $x, x'$ we have without loss of generality that $\varepsilon(x, x') \leq q\varepsilon_1 + \varepsilon_2$ and $\varepsilon(x', x) \leq (1 - q)\varepsilon_1$. Therefore it must be $(\varepsilon_1 + \varepsilon_2)$-range-bounded and applying Proposition 2 gives the zCDP guarantees.

$\square$

## C.3 Extension to multiple quantile estimation

As previously established, the framework in Kaplan et al. (2022) is agnostic to the single quantile estimation method. However their proof is for the *add-subtract* neighbors, although they note that it can be extended easily to the *swap* neighbors. We also discuss here how it can be extended for completeness.

For the *swap* neighboring definition, at each level of the recursive partitioning scheme either two partitions differ under the *add-subtract* definition, or one partition differs under the *swap* definition. Applying their framework to our algorithm, we will instead compute all the thresholds in advance of the splitting. Accordingly these thresholds will remain fixed. If the thresholds are fixed then we can actually further improve our privacy bounds for the *add-subtract* definition. Once again we will use our more general characterization from Section B.1.

**Lemma C.2.** *Given the stream of queries $f_i(x) = |\{x_j \in x | x_j - \ell + 1 < \beta^i\}|$ with sensitivity $\Delta = 1$, for any neighboring datasets $x, x'$ under Definition C.1 we have $\varepsilon(x, x') \leq \max\{\varepsilon_1, \varepsilon_2\}$*

*Proof.* Without loss of generality, assume that $x$ has one more individuals data, so $x \in \mathbb{R}^n$ and $x' \in \mathbb{R}^{n-1}$. By construction we must have $f_i(x) - f_i(x') \in \{0, 1\}$ because $x$ has an additional datapoint. Furthermore, because the thresholds are increasing, if $f_k(x) - f_k(x') = 0$ then $f_i(x) - f_i(x') = 0$ for all $i < k$. Thus the case of $f_k(x) - f_k(x') = 0$ is easy.

Instead consider $f_k(x) - f_k(x') = 1$. We know $\Delta_k(x, x') = 0$ so $\Delta_k(x, x') - (f_k(x') - f_k(x)) = 1$, so $\varepsilon(x, x') \leq \varepsilon_2$. Further we must have $\Delta_k(x', x) \leq 1$ and $\Delta_k(x', x) - (f_k(x) - f_k(x')) = 0$, so $\varepsilon(x', x) \leq \varepsilon_1$. Combining these implies $\varepsilon(x', x) \leq \max\{\varepsilon_1, \varepsilon_2\}$ for any neighboring datasets.

$\square$

Applying these bounds we see that applying the framework of Kaplan et al. (2022) to the swap definition either leads to one composition of $(\varepsilon_1 + \varepsilon_2)$-DP for the *swap* definition or two compositions of $\max\{\varepsilon_1, \varepsilon_2\}$ for the *add-subtract* definition at each level. Setting $\varepsilon_1 = \varepsilon_2$ we then have that the privacy cost of computing $m$ quantiles with this framework will be equivalent to $\log(m + 1)$ compositions of $(\varepsilon_1 + \varepsilon_2)$-DP.

## C.4 Applying permute-and-flip framework to EMQ

It would be an interesting future direction to see if the EMQ method could also instead effectively utilize the permute-and-flip framework from McKenna and Sheldon (2020) that was shown to improve accuracy. While this framework is equivalent to adding Expo noise for report noisy max, the quantile problem requires considering an infinite domain, which makes the Expo noise addition procedure impossible. Using this alternate, but equivalent Ding et al. (2021), approach can make this possible. If we look at Algorithm 3 from McKenna and Sheldon (2020) we could also extend this to drawing a single point uniformly in the interval, then drawing the Bernoulli and removing it from the interval. However due to the continuous nature of the interval, it's unlikely that performing this sampling without replacement will have any noticeable improvement over sampling with replacement which is equivalent to the exponential mechanism. Furthermore, we cannot run this framework as efficiently because there are no corresponding nice closed form expressions compared to the exponential mechanism. Ideally, we would modify this approach to draw each interval proportional to it's length, then draw the Bernoulli and remove the interval. However it is critical to note that this does not give an identical distribution because the sampling is done without replacement. Accordingly this simple modification would not work, but there could potentially be other effective changes to fit this framework that we leave to future work.

Furthermore, we note that adding Expo noise for report noisy max does not achieve the range-bounded property that the exponential mechanism enjoys Durfee and Rogers (2019). So the composition improvements for zCDP from Proposition 2 could not be applied.

# D Further Experiment Details

In this section we provide some follow-up details from our empirical evaluation.

## D.1 Data histograms

We provide histograms of our datasets for better understanding in Figure 3.

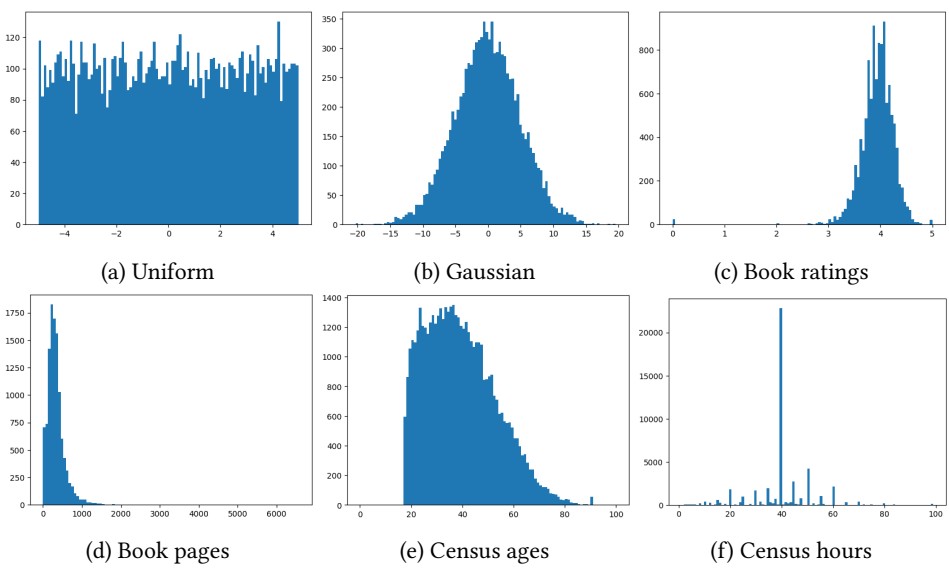

|  |  |  |
|---|---|---|
| (a) Uniform | (b) Gaussian | (c) Book ratings |
| (d) Book pages | (e) Census ages | (f) Census hours |

Figure 3: Histograms for each of our datasets.

## D.2 Parameter tuning

We kept our $\beta$ parameter fixed in all experiments for consistency but also to make our method data agnostic. However, our choice was aggressively small in order to achieve higher precision in the inner quantile estimation comparison in Section 5.2. This choice was still highly resilient to changes in $\varepsilon$ for our sum experiments as we see our error only scaled proportional to the increase in noise. But for more significant decreases in $\varepsilon$ or in the data size, i.e. the conditions under which all private algorithms suffer substantial accuracy loss, this choice of $\beta$ could be too small. Those settings imply that the noise added is larger and the distance between queries and threshold shrinks, so our method is more likely to terminate earlier than desired. Smaller $\beta$ values will then intensify this issue as the candidate values increase more slowly. For the clipping application, we generally think using a value of $\beta = 1.01$ would be a more stable choice. In fact, some preliminary testing shows that this setting actually improves our results in Table 1. Furthermore, increasing $\beta$ will also reduce the number of queries and thus the computational cost. We leave to future work a more thorough analysis of this parameter to determine a good default value that performs well agnostic to the input data. Additionally, all our methods and proofs only require an increasing sequence of candidate values and it's possible that other potential sequences would be even more effective. For example, if tight upper and lower bounds are known on the data, such as within the recursive computation of multiple quantiles, then it likely makes more sense to simply uniformly partition the interval and check in increasing order. But we leave more consideration upon this to future work as well.

Our choice of $q = 0.99$ in the differentially private sum experiments was also to maintain consistency with the choices for the previous method but also to keep variance of the estimates lower. This will create some negative bias as we then expect to clip the data. As we can see in our illustrative example Figure 2, the PDF will exponentially decrease once we pass the true quantile value, but will do so less sharply once all the queries have value $n$. Accordingly, setting $q = 1.0$ would add slightly more variance to the estimation but initial testing showed improvement in error. However, if the user would prefer slightly higher variance to avoid negative bias, then setting the threshold at $n$ or even $n + 1/\varepsilon$, would make it far more likely that the process terminates with a value slightly above the maximum. This is particularly useful for heavy-tailed data, where clipping at the 99th percentile can have an out-sized impact on the bias. We leave a more rigorous examination of these bias-variance tradeoffs for good default settings to future work.

### D.3 Implementation code

For ease of implementation, we provide some simple python code to run our method with access to the respective `Noise` generator of the users choosing. In our experiments we used exponential noise from the numpy library.

```python
def unboundedQuantile(data, l, b, q, eps_1,eps_2):
  d = defaultdict(int)
  for x in data:
    i = math.log(x-l+1,b) // 1
    d[i] += 1

  t = q * len(data) + noise(1/eps_1)
  cur, i = 0, 0
  while True:
    cur += d[i]
    i += 1
    if cur + noise(1/eps_2) > t:
      break
  return b**i - l + 1
```