# OpenReview forum: "Unbounded Differentially Private Quantile and Maximum Estimation"
_NeurIPS.cc/2023/Conference — NeurIPS 2023 poster_

### Official Review · Reviewer_QwHj · 2023-07-01

**Soundness:** 3 good
**Presentation:** 3 good
**Contribution:** 2 fair
**Rating:** 6
**Confidence:** 3

**Summary:**

The paper considers the problem of differentially private computation of quantiles for the data. The core idea is to leverage the ‘AboveThreshold’ algorithm  as a subroutine to guess the desired item value such that the rank of the item value is about qn for target quantile q. Authors also discuss the application of this method for mean and sum estimation where the proposed algorithm can help clipping the item values.Experimental results indicate improvement for inner quantiles over Page, Age, Workhours dataset.

**Strengths:**

Quantile estimation is very important problem with real-world applications. This paper presents a straightforward application Abovethreshold differential privacy techniques. For applications, the simplicity in algorithms is greatly valued. The paper is well written and very accessible for the general audience.

**Weaknesses:**

 One major concern is the requirement of O(n) passes through the dataset. With a large volume of data, n passes seem to be formidable. The experimental section can be strengthened by using more data points and considering query latency.

Related works on quantile approximation are missing.
     Gillenwater, Jennifer, Matthew Joseph, and Alex Kulesza. "Differentially private quantiles."  They present a DP-algorithm with exponential mechanism that estimates all m quantiles in $O(mnlog(n)+m^2n)$ time. Zhao, Fuheng, et al. "Differentially private linear sketches: Efficient implementations and applications."  && Pagh, Rasmus, and Mikkel Thorup. "Improved utility analysis of private countsketch." They present a DP-algorithm using count sketch to estimate all quantiles in one pass with gaussian mechanism.

**Questions:**

1) Can the algorithm improve to O(logN) pass over the data with something like exponential search?

2) 1000 data points in the experiments are very small. Also maybe consider including some results on the query latency. For example the time to find the median.


**Limitations:**

No potential negative societal impact.

---

> ### Author Rebuttal · Authors · 2023-08-08
>
> We thank the reviewer for their feedback and apologize for not stating the runtime of our result more clearly. To be very clear, our algorithm only requires ONE pass through the data, so the runtime is O(n) in the data size, and does not require sorted data. We point the reviewer to section 4.2 for more details and section D.3 in the supplementary material for the corresponding code which is quite simple.  Furthermore this pass through the data is easily parallelizable which allows it to scale even better for large datasets. If we also apply the splitting framework (see section 4.3) for multiple quantiles, then this will only require log(m) passes through the data taking O(log(m) n) time in the data size.
>
> We definitely see how our wording was confusing, but we hope this should clear up most of the reviewer's questions and concerns. We also note that we do reference and discuss "Differentially private quantiles" from Gillenwater et al.  multiple times (and using 1000 data points was a result of copying their experimental setup). We did not include explicit runtime comparison, though as you can see ours will do better asymptotically in the data size. We also thank the reviewer for pointing out the missing countsketch references that we will be sure to add with discussion.

---

> > ### Comment · Reviewer_QwHj · 2023-08-18
> >
> > Thanks a lot for providing more information on the work such as revised proof and detailed responses to reviewers. I slightly raised my score.

---

> > > ### Author Response · Authors · 2023-08-18
> > >
> > > We sincerely thank the reviewer for considering our rebuttals and raising their score! We apologize again for not clearly stating our runtime, which is asymptotically equivalent in the data size to quickselect, the fastest non-private quantile algorithm. We appreciate the reviewer bringing this confusion to our attention and we will be sure to make the efficiency of our algorithm clear in future versions of our work.

---

### Official Review · Reviewer_VnsR · 2023-07-04

**Soundness:** 1 poor
**Presentation:** 2 fair
**Contribution:** 2 fair
**Rating:** 4
**Confidence:** 2

**Summary:**

The paper tries to improve upon Sparse Vector Techniques for differential privacy to obtain better estimates of the maximum and quantiles.

**Strengths:**

The paper covers an important problem, finding a differentially private estimate of the max. This  is often needed in one of the most common aggregations, i.e. sums, to clip and set a sensitivity.

 It promises less noise for the same privacy through an improvement to the sparse vector technique.

I would be interested in reading it again if the proofs were correct. However,...

**Weaknesses:**

I unfortunately cannot trust the results in the paper due to bad proofs.

The proof for Lemma B.1 is incorrect though the conclusion may not be. For example, the statement in Line 557 of the supplement is incorrect. The statement says you will get the same answer _with the exact same noise_. You’ve fixed the randomness and there is no remaining randomness for the probabilities. You end up using v’ \neq v and v_k’ \neq v_k later.

Furthermore, injectivity of the mapping v, v_k -> v’, v_k’ is not enough to show that the probabilities P(AboveThreshold(x) = O) and P(AboveThreshold(x’) = O) are the same for some output O. For example x -> x/2 is an injective mapping from [0,1]. But for a uniform distribution, the probability of the image is ½, not 1. In this case, the mapping happens to be a shift, so the Jacobian of the map is the identity, which actually is enough since the determinant is 1. But the current proof is too much of a mess to know if I can trust the theoretical claims in the paper.

Please fix the reference in Lemma 3.1. The Thm 4, 5 in the VLDB version are show that algorithms are _not_ DP.

**Questions:**

Please fix the proofs!

It is hard to tease out your contributions from the contributions section. Most of the contributions section belongs in the introduction. The contributions should be a succinct description of what _you_ contributed, not about where it can be used.

**Limitations:**

Yes

---

> ### Author Rebuttal · Authors · 2023-08-08
>
> We thank the reviewer for pointing out the incorrect reference (should be Thm 2 and 3 for the VLDB version) in Lemma 3.1. Our reference in line 618 is also for Lemma 3.4 in the arxiv version of Cesar and Rogers and needs to be updated.
>
> We felt that the flow and clarity of the introduction was best achieved through our format but should have used "Our techniques" as that subsection header instead and apologize for the misnomer. For a concise review of our results/contributions, we point the reviewer to the abstract.
>
> $ $
>
> $\textbf{Clarifying the notation confusion which led to the reviewer's main issue:}$
>
>
> We apologize for the notational confusion in our proof of Lemma B.1 which led to the main issue for the reviewer. In line 557, when we fix the randomness of $\nu_{i<k}$ this denotes all $\nu_i$ such that $i<k$. Hence there is still randomness from $\nu_k$ and $\nu$, which is what the probabilities are over, and why we later have $\nu \neq \nu'$ and $\nu_k \neq \nu_k'$. We hoped this would be clear from context and strongly suggestive notation but will be sure to clarify in the future. We also invite the reviewer to see the proof of Claim 3.9 in The Algorithmic Foundations of Differential Privacy by Dwork and Roth for a similar style of proof (that inspired our approach) which may further help understanding.
>
> $\textbf{Updating the proof with an easy addition that addresses an important technicality}$
>
> We greatly appreciate the reviewer pointing out that our change in variable is a shift so the Jacobian is the identity matrix with determinant of 1, which is easily added but an important technicality. We'll be certain to anonymously acknowledge the reviewer for this contribution. We apologize for this oversight and can add this update and better organize the rest of the proof after line 557 as:
>
> $ $
>
> Let $\Delta_1 = \max(0, \tau'-\tau)$ and $\Delta_2 = \max(0, \Delta_1 - (f_k(x') - f_k(x)))$ and let $\nu' = \nu + \Delta_1$ and $\nu_k' = \nu_k + \Delta_2$. It suffices to show that for every pair of draws $\nu, \nu_k$ that satisfies $\tau < T + \nu < f_k(x) + \nu_k$, then the corresponding $\nu', \nu_k'$ are (1) non-negative, (2) satisfy $\tau' < T + \nu' < f_k(x') + \nu_k'$ and (3)
>
> $
> p_{expo}(\nu;\Delta/\epsilon_1) \cdot p_{expo}(\nu_k;\Delta/\epsilon_2) \leq \exp({\epsilon}(x,x')) p_{expo}(\nu';\Delta/\epsilon_1) \cdot p_{expo}(\nu_k';\Delta/\epsilon_2)
> $
>
> We note that condition (3) does not require any other scalar on the RHS as our change of variable is a shift for each so the Jacobian of the mapping is the identity matrix with determinant of 1.
>
>
> Condition (1) follows from the fact that $\nu,\nu_k$ are non-negative because they're drawn from the exponential distribution, and $\Delta_1, \Delta_2$ are non-negative by construction.
>
> For condition (2), if $\tau < T + \nu$ we must have $\tau' < T + \nu + \Delta_1$ by definition of $\Delta_1$. Similarly, if $T + \nu < f_k(x) + \nu_k$ then $T + \nu + \Delta_1 < f_k(x') + \nu_k + \Delta_2$ because $\Delta_2 \geq \Delta_1 - (f_k(x') - f_k(x))$.
>
> Finally, for condition (3), the PDF of the exponential distribution implies $p_{expo}(\nu;\Delta/\epsilon_1) \leq \exp(\frac{\epsilon_1 \Delta_1}{\Delta}) p_{expo}(\nu';\Delta/\epsilon_1)$ and $p_{expo}(\nu_k;\Delta/\epsilon_2) \leq \exp(\frac{\epsilon_2 \Delta_2}{\Delta}) p_{expo}(\nu_k';\Delta/\epsilon_2)$. Due to the fixing of randomness of $\nu_{i<k}$, we have that $\tau' - \tau \leq \max_{i < k} (f_i(x') - f_i(x))$ so $\Delta_1 \leq \Delta_k(x,x')$ from Definition B.1 and $\Delta_2 \leq \max (0, \Delta_k(x,x') -(f_k(x') - f_k(x)))$ which implies condition (3). Q.E.D.
>
> $ $
>
>
>
>
> We also mention in line 569 that this proof applies to Laplace noise. The only change is that condition (1) is no longer required as Laplace noise can be negative, and for condition (3) the inequalities hold identically for the laplace PDF.
>
> $ $
>
> $\textbf{Preemptively giving other further proof details and updates for better clarity}$
>
>
> Preemptively we feel the reviewer will also want more detail for Lemma B.2 although it's an immediate application of Lemma 4.2 (which is a folklore result) from Durfee and Rogers once the notation is translated. Specifically, for their $M^k_{gumbel}$ defined on page 11 (of their arxiv version), we'll let the output set be $Y = (i : i \leq k ) \cup \tau$ where $\tau$ represents the threshold. Further, let $q(x, i) = f_i(x)$ and $q(x,\tau) = T$. By construction the probability that ${M}_{gumbel}^\ell(q(x)) = (k, \tau)$, where $\ell=2$, is equivalent to the probability of above threshold (with Gumbel noise) returning $(\bot^{k-1},\top)$. Even more specifically, both can be identically written in the integral form at the beginning of their proof of Lemma 4.2 (top of page 36 of their arxiv version) but translated to the outcome of $(k ,\tau)$. If we then look at the last line of the proof of Lemma 4.2 (page 36 of their arxiv version) and plug in $\epsilon_q$ defined earlier in their proof, we see that this lemma immediately implies our desired result.
>
> $ $
>
>
> Upon detailed re-examination of all our proofs we also discovered that Lemma B.3 needs minor changes. In particular, the relaxation in line 589 must be removed. This does not affect any of the results in the main body of our paper, and only requires we remove the second claim in Corollary C.1 which is not used anywhere else in our results. We apologize for our embarrassing sloppiness and thank the reviewer for forcing a thorough re-examination upon us to improve the clarity of our work. Updating the proof only requires re-setting $\Delta_1 = \max(0,\Delta_1)$ and $\Delta'_1 = \max(0, \Delta'_1)$, but otherwise follows the exact same logic. We also found a typo in the proof of Lemma C.1 at the beginning of line 720, which should read $\Delta_k$ instead of $\Delta_i$. We'll be more than happy to immediately provide the edited and more detailed proof of Lemma B.3 in this thread, but it would exceed the character limit in this response.

---

> > ### Comment · Reviewer_VnsR · 2023-08-20
> >
> > Thank you for your responses. While I feel this is better addressed in another review cycle, I've slightly increased my score, and lowered my confidence. I hope you found the review helpful in improving the paper.

---

> > > ### Author Response · Authors · 2023-08-20
> > >
> > > We are incredibly grateful to the reviewer for pointing out the easily added but important technicality, which has helped improve the clarity of our proofs. We also appreciate the reviewer revisiting their score and confidence!
> > >
> > > While we are very embarrassed to have missed this technicality in the first place, we strongly feel that these types of minor technical updates and clarifications are quite common within the rebuttal period, and are in fact the main purpose of the rebuttal period. Our updates are only minor revisions where the underlying logic remained unchanged and have no effect on the main body of the paper. The notion that our proofs were fundamentally flawed entirely stemmed from our not specifying that $\nu_{i<k}$ denoted all $\nu_i$ such that $i<k$. We apologize for this omission, but hope that the entirety of our work will not receive biased judgement that major revisions were needed as a result of such a tiny oversight.

---

### Official Review · Reviewer_nQ9t · 2023-07-07

**Soundness:** 3 good
**Presentation:** 4 excellent
**Contribution:** 3 good
**Rating:** 7
**Confidence:** 3

**Summary:**

The authors propose a differentially private algorithm for differentially private quantile estimation that can handle unbounded data setting. The authors revisited the AboveThreshold subroutine in Sparse Vector Technique and proved new privacy guarantee results. The authors can verify the asymptotic runtime complexity and demonstrate the utility through experiments. The method can have successful application to differentially private sum estimation.

**Strengths:**

**Originality** The proposed method for finding private quantile is a creative use of an existing method for the application to a new domain. The authors proved new privacy results for the existing method.

**Quality**  The method can solve the problem. The theoretical analysis can verify the privacy and complexity of the proposed method. The authors can demonstrate the accuracy of the method through numerical studies.

**Clarity** The problem is well-defined and clearly stated. The method can be clearly understood.

**Significance** The problem being solved is important and as verified by the experiment the proposed method can improve accuracy for real-world data . The new privacy analysis to the Sparse Vector Technique can be of independent interest.

**Weaknesses:**

The authors do not provide theoretical utility guarantee; though the experiment results look good.



**Questions:**

1. You provide theoretical runtime complexity analysis. It would be of interest to compare the runtime in experiments to verify your asymptotic analysis.

2. From a quick search, there seems to be little literature in quantile estimation that addresses the unbounded setting.  It would be great to highlight this by citing some papers that can handle unbounded data in Introduction/Related Work.  You briefly mentioned this in your contributions, saying “commonly used algorithms can apply loose bounds to ensure data is contained within” and analyzing their drawbacks, but no citation is included.

**Limitations:**

The proposed method works worse than the existing EMQ method for synthetic data. Can you offer any suggestions on how to improve your method’s accuracy for these well-behaving datasets?

---

> ### Author Rebuttal · Authors · 2023-08-08
>
> We thank the reviewer for taking the time to review and give detailed feedback on our work.
>
> Response to questions:
>
> 1. We felt that verification of runtime asymptotics was less necessary given the simplicity of the code (see Section D.3 in the supplemental file), but agree that experimental runtime verification would be a good addition particularly for larger datasets.
>
> 2. This is a great suggestion and we'll look for work that handles unbounded data for non-quantile algorithms. Our mention of applying "loose bounds" (line 61) was specifically with respect to quantile algorithms, and to our knowledge none of these works explicitly address this or consider the drawbacks. Instead the bounds are just assumed to be reasonably tight. Our intention with that statement (line 61) was to give credit to these works that technically they can handle unbounded data (in practice) by using incredibly loose bounds even though it's never actually stated in their work. But as seen in our results for maximal quantile estimation, even somewhat loose bounds can have significant drawbacks on accuracy.
>
>
> Response to limitation question:
>
> This is a really good question and we've been developing some further intuition for the inner quantiles (but concrete theoretical claims appear extremely difficult). The general intuition we've additionally found is that the more symmetric the data is in a "small" neighborhood around the quantile, the better EMQ performs comparatively. Most real world data does not have this symmetry hence our UQE performing better. This also implies that our UQE would perform better (and we have empirically verified) for synthetic data generated from the exponential distribution or poisson distribution (with small parameter). We also note that this is for inner quantiles, and for the maximal quantiles (and loose upper bounds) our UQE also significantly outperforms EMQ on synthetic data.

---

> > ### Comment · Reviewer_nQ9t · 2023-08-15
> >
> > Dear authors,
> >
> > Thanks for your responses. I recommend you include your response to the limitation question (and/or) some supporting experiments somewhere in the main text and supplementary of the paper. I would be happy to see if you have any new results that support your discussion.
> >
> > I don’t think you have addressed my concerns in **Weaknesses**. Do you have a utility guarantee similar to Theorem 3.3 of your cited work *Kaplan et al. (2022)* or in whatever error metrics that tells how your estimations differ from the true quantiles? I believe this is needed to make the paper more theoretically complete.
> >
> > Following your discussion with reviewer VnsR, could you please also send an updated proof as a stand-alone pdf? I would be happy to check the correctness and provide some comments if you can provide it before the rebuttal ends.
> >
> > Thanks again

---

> > > ### Author Response · Authors · 2023-08-15
> > >
> > > Thanks again to the reviewer for all the detailed suggestions and feedback! We also greatly appreciate the reviewers willingness to look over the minor clarification and updates to our proof. According to the rules, only figures and tables can be included in the pdf, so we've contacted the AC about making an exception. The proof updates can also be found in our rebuttal but we can definitely understand the pdf formatting is easier to review and we'll hopefully be able to send a pdf of the proof very shortly.
> > >
> > >
> > > We will also be sure to include our limitations discussion in future versions and appreciate the reviewers suggestions. We do not have sufficient theoretical evidence to make concrete claims, but we will certainly add our conjectured intuition and some supporting experiments that may be highly useful to practitioners and future work.
> > >
> > >
> > >
> > > $\textbf{Regarding the theoretical utility guarantees}$
> > >
> > > We apologize for failing to address the weakness mentioned by the reviewer. We can improve the $(\alpha, \beta)$ guarantees of SVT in Theorem 3.24 of [1] (and cited in [2]) with our new results using Gumbel or Exponential noise. We will be sure to include these improved guarantees in future versions of our work and thank to reviewer for this suggestion.
> > >
> > > In particular, if we run AboveThreshold with Gumbel noise and $\epsilon_1 = \epsilon_2 = \epsilon/3$ (which gives $\epsilon$-DP for AboveThreshold) then we can improve the guarantees of Theorem 3.24 in [1] to $ \alpha = 3 ( \log(k) + \log(1/\beta)) / \epsilon$. In the theorem and algorithm of [1] it is assumed that the sensitivity $\Delta=1$ which we will also apply for simplicity but can be generalized.
> > >
> > > $ $
> > >
> > > Proof sketch that we'd be happy to elaborate upon:
> > >
> > > According to Definition 3.9 of [1], we want to show that the probability of halting before query k is at most $\beta$. Let $\nu$ be the noise added to the threshold hold, let $\nu_i$ be the noise added to query $f_i$, and let $\tau = max_{i\leq k} f_i(x) + \nu_i$. We know AboveThreshold will not halt before $k$ if the noisy threshold is above all noisy queries $i \leq k$. So we want to show that $Pr(T + \nu > \tau) \geq 1 - \beta$. It is known that adding Gumbel noise and selecting the greatest noisy value is equivalent to the exponential mechanism (see Lemma 4.2 of [3] for a proof). Therefore, given that our gumbel noise parameter is $3 / \epsilon$ we have $Pr(T + \nu > \tau) = \exp( \epsilon T / 3) / (\exp( \epsilon T / 3) + \sum \exp(\epsilon f_i(x) / 3))$. We then apply the assumption from Theorem 3.24 of [1] that $f_i(x) \leq T - \alpha$ for all $i$ and our goal is to show  $\exp( \epsilon T / 3) / (\exp( \epsilon T / 3) + k \exp(\epsilon (T- \alpha) / 3)) \ \geq 1 - \beta$. Some straightforward algebraic manipulation then shows that $ \alpha = 3 ( \log(k) + \log(1/\beta)) / \epsilon$ satisfies this inequality.
> > >
> > > $ $
> > >
> > >
> > > The leading constant for this $\alpha$ guarantee can also be improved identically to 2 for the monotonic setting as we can set $\epsilon_1 = \epsilon_2 = \epsilon / 2$ to achieve $\epsilon$-DP. Further we can show the same improvement for adding exponential noise which we would be happy to provide a proof as well. We think that these results could also likely be applied to give theoretical guarantees for the quantile algorithm, but leave such analysis to future work. We would like to note that our empirical analysis was of the exact same datasets and experimental setup as in [4] so as to give fair practical comparison.
> > >
> > > $ $
> > >
> > > [1] "The Algorithmic Foundations of Differential Privacy",  Dwork and Roth
> > >
> > > [2] "Understanding the Sparse Vector Technique for Differential Privacy" Lyu, Su, and Li
> > >
> > > [3] "Practical Differentially Private Top-k Selection with Pay-what-you-get Composition" Durfee and Rogers
> > >
> > > [4] "Differentially Private Approximate Quantiles", Kaplan, Schnapp, Stemmer

---

### Official Review · Reviewer_Qz1m · 2023-07-07

**Soundness:** 4 excellent
**Presentation:** 4 excellent
**Contribution:** 3 good
**Rating:** 7
**Confidence:** 4

**Summary:**

The paper considers the problem of computing the quantile q of a dataset with differential privacy. The prior standard technique for this problem was the exponential mechanism with the loss of an output T being the difference between the number of examples less than T and qn. An issue with the exponential mechanism (and most mechanisms for quantile estimation) is that it requires the user to bound the range of possible values in the dataset. The authors propose an alternate algorithm which does not require a prior bound, and takes advantage of the AboveThreshold algorithm. They propose using the AboveThreshold on queries of the form "how many examples in the dataset are less than T?", where T is exponentially increasing, and return the value T that AboveThreshold reports has more than qn elements less than T. As part of the analysis, the authors give an improved analysis of AboveThreshold, showing it has improved privacy guarantees for monotonic queries when using Exponential or Gumbel noise; previously, this was shown for Laplace noise. The Exponential distribution has lower variance, so this allows them to get a lower-variance estimate without sacrificing privacy. The authors perform experiments to compare to the exponential mechanism, as well as the aggregate tree method. On synthetic datasets, the exponential mechanism performs slightly better, but otherwise the authors' result is better for quantile estimation. For sum estimation, when using parameters agnostic to the dataset, the authors' algorithm by far outperforms the baselines, due to not relying on a good prior range estimate.

**Strengths:**

Overall I enjoyed reading the paper and feel it makes a substantial contribution to the field. Private quantile selection is an extremely important problem with a number of applications, including any setting where we wish to compute a private sum. The problem of needing a bounded range for these problems has always been a troublesome part of DP research and application, and having a seemingly robust approach that at least partially gets around this issue is thus of significant impact. The main idea in the paper is simple to describe, which makes it more likely to see adoption and further development in follow-up works, but at the same time requires new analyses to achieve competitive privacy-utility tradeoffs. Finally, I felt the paper was overall a pretty easy read, in part due to the elegance of the algorithmic ideas but also the quality and clarity of the writing.

**Weaknesses:**

I think the only major weakness is that one still needs to choose beta appropriately. The authors mention this in the discussion in Appendix D, but this discussion is somewhat incomplete. Still, I believe that beta is a much easier parameter for a practitioner to choose appropriately than the range of the dataset, especially since the experiments show beta is somewhat dataset-agnostic, so this is a minor weakness.

**Questions:**

Intuitively it seems like smaller beta is more likely to underestimate the quantile and larger beta is more likely to overestimate, is this correct? Also, for e.g. a bimodal dataset such as the Gaussian mixture 0.5 * N(0, 1) + 0.5 * N(Delta, 1), Delta >> 1, for estimating the 75th percentile, how would you expect UQE to compare to EMQ? (with a good or bad range estimate)

**Limitations:**

Yes, in the appendix the authors discuss the main limitation/weakness. N/A on negative societal impact.

---

> ### Author Rebuttal · Authors · 2023-08-08
>
> We thank the reviewer for thoroughly reading our work, giving extensive and detailed feedback, and are very pleased to hear that they enjoyed doing so.
>
>
> As the reviewer mentioned we largely left a more detailed examination of beta settings to future work, partly to demonstrate that a reasonable guess of beta without any prior would still perform very well. Since the reviewer has shown further interest, which we greatly appreciate, we will mention that the general rule of thumb we have been further considering is that beta should be inversely proportional to the data size and epsilon. This also relates to the reviewer's intuition (which is spot-on) but extends it further.  Larger beta, say $\beta=1.1$, will also lead to fewer guesses close to the true quantile. In fact, even with $\epsilon = \infty$ we could still only expect to be within about 5% of the true quantile. However, for smaller datasets and small epsilon, i.e. the situation under which all differentially private algorithms struggle with accuracy, we could only hope to have coarser estimates and setting beta higher will avoid significant underestimation that the reviewer's intuition caught. We note that we did find our experiments to show further robustness with respect to beta as we re-ran them afterwards with $\beta = 1.01$ and found similar (if not greater) improvements for the real world data. As such, we believe a good default setting would not require too much specificity and a simple heuristic function that sets beta inversely proportional to the data size and epsilon should suffice. But this still needs further development. For now a selection of $\beta \in [1.001, 1.01]$ seems to be a well-performing and reasonably robust default.
>
>
>
>
> Response to questions:
>
> 1. The reviewer is spot-on with their intuition that smaller beta more likely leads to underestimation and larger beta more likely leads to overestimation.
>
> 2. This is a really interesting question and we've been developing some further intuition for the inner quantiles (but concrete theoretical claims appear extremely difficult). The general intuition we've found so far is that the more symmetric the data is in a "small" neighborhood around the quantile, the better EMQ performs comparatively. Most real world data does not have this symmetry hence our UQE performing better. For the example the reviewer gives, the significant distance between the Gaussians compared to the variance would mean we would expect the 75th percentile to be symmetric and EMQ to likely perform better. But with more overlap and differing variances this symmetry would disappear and our intuition is that UQE would often perform better.

---

> > ### Comment · Reviewer_Qz1m · 2023-08-14
> >
> > Thank you for the response, and the detailed answers to my questions. As of now I'm still of the opinion the paper deserves acceptance, as it seems the concerns of reviewer VnsR are likely to be addressed during the rebuttal process (although I will of course plan to stay up-to-date on that thread to confirm this)

---

> > > ### Author Response · Authors · 2023-08-14
> > >
> > > We again thank the reviewer for such a detailed review! We'll be more than happy to answer any questions the reviewer may ultimately have on the thread clarifying and updating the proofs. We want to make sure to provide any and all helpful follow-up details so that hopefully all the reviewers are in agreement on the correctness of our theoretical work.

---

### Comment · Area_Chair_yBWF · 2023-08-17
**Anonymous link to updated proof.**

I have uploaded a PDF of the author's revised proof (discussed in their rebuttal) to the following anonymous link: https://drive.google.com/file/d/1-tG7AwFugTTCRHpObidWzAtCbJdTOSBK/view?usp=sharing

Please let me know if you have any trouble accessing the file.

Thanks,
Your AC

---

### Decision · Program_Chairs · 2023-09-21

**Decision:**

Accept (poster)

**Comment:**

Reviewers Qz1m, nQ9t, and QwHj are all in favor of accepting the paper, arguing that the proposed unbounded quantile mechanism is simple with interesting and novel analysis, applies standard DP mechanisms creatively, and that the underlying problem is important. On the other hand, reviewer VnsR argues that some proofs are incorrect.

My understanding is that the two concrete concerns raised by VnsR have been addressed by the authors. In particular, regarding line 557, I think it is clear that the authors are only fixing the values of $v_1, \dots, v_{k-1}$ and not $v$ or $v_k$. It also looks like the authors acknowledge the correction required in their change of variables (which does not affect the calculation, since the determinant of the Jacobian is 1). While I agree that these are important corrections, it seems like these are relatively minor errors that can be fixed in the camera ready. During the reviewer discussion period I asked if there were any further correctness concerns and reviewer VnsR did not respond.

With these corrections, I believe the paper is above the bar for NeurIPS.